# The pioneer factor OCT4 requires the chromatin remodeller BRG1 to support gene regulatory element function in mouse embryonic stem cells

Hamish W King, Robert J Klose*

Department of Biochemistry, University of Oxford, Oxford, United Kingdom

**Abstract** Pioneer transcription factors recognise and bind their target sequences in inaccessible chromatin to establish new transcriptional networks throughout development and cellular reprogramming. During this process, pioneer factors establish an accessible chromatin state to facilitate additional transcription factor binding, yet it remains unclear how different pioneer factors achieve this. Here, we discover that the pluripotency-associated pioneer factor OCT4 binds chromatin to shape accessibility, transcription factor co-binding, and regulatory element function in mouse embryonic stem cells. Chromatin accessibility at OCT4-bound sites requires the chromatin remodeller BRG1, which is recruited to these sites by OCT4 to support additional transcription factor binding and expression of the pluripotency-associated transcriptome. Furthermore, the requirement for BRG1 in shaping OCT4 binding reflects how these target sites are used during cellular reprogramming and early mouse development. Together this reveals a distinct requirement for a chromatin remodeller in promoting the activity of the pioneer factor OCT4 and regulating the pluripotency network.

*For correspondence: rob.klose@bioch.ox.ac.uk

Competing interests: The authors declare that no competing interests exist.

## Introduction

Transcription factors read DNA-encoded information to control gene transcription and define the complement of proteins a cell can produce. They achieve this by physically binding specific DNA sequence motifs in gene regulatory elements that ultimately control how RNA polymerases engage with and transcribe genes (*Spitz and Furlong, 2012*; *Voss and Hager, 2014*). In prokaryotes, transcription factors tend to function as self-contained units that recognize extended DNA sequences with high specificity and affinity (*Wunderlich and Mirny, 2009*). Therefore, simple sequence-based principles appear to dictate their binding in the genome. In contrast, transcription factors in higher eukaryotes recognize shorter DNA sequences and often interface with other DNA-binding transcription factors in a combinatorial manner to achieve affinity and specificity (*Wunderlich and Mirny, 2009*; *Villar et al., 2014*). In addition to the widespread requirement for combinatorial DNA binding, eukaryotic transcription factors are also confronted with the added complexity that eukaryotic DNA is wrapped around histones to form nucleosomes and chromatin (*Kornberg and Lorch, 1999*). The association of DNA with nucleosomes can therefore compete with DNA-binding transcription factors, limiting their ability to engage with the genome (*Spitz and Furlong, 2012*; *Voss and Hager, 2014*). Indeed, in many cases transcription factor binding appears to require pre-existing chromatin accessibility (*Guertin and Lis, 2010*; *Biddie et al., 2011*; *John et al., 2011*).

This chromatin barrier therefore poses a significant challenge in establishing new gene regulatory elements when cells differentiate during development. Therefore, to overcome this obstacle, cells have evolved a specialised set of transcription factors that are able to recognise their cognate motifs

**eLife digest** All cells in your body contain the same genetic information in the form of genes encoded within DNA. Yet, cells use this information in different ways so that the activities of individual genes within that DNA can vary from cell to cell. This allows identical cells to become different to each other and to adapt to changing circumstances.

A group of proteins called transcription factors control the activity of certain genes by binding to specific sites on DNA. However, this isn't a straightforward process because DNA in human and other animal cells is usually associated with structures called nucleosomes that can block access to the DNA. Pioneer transcription factors, such as OCT4, are a specific group of transcription factors that can attach to DNA in spite of the nucleosomes, but it's not clear how this is possible. Once pioneer transcription factors attach to DNA they can help other transcription factors to bind alongside them.

King et al. studied OCT4 in stem cells from mouse embryos to investigate how it is able to act as a pioneer transcription factor and control gene activity. The experiments show that several other transcription factors lose the ability to bind to DNA when OCT4 is absent. This leads to widespread changes in gene activity in the cells, which seems to be due to other transcription factors being unable to get past the nucleosomes to attach to the DNA.

Further experiments showed that OCT4 needs a protein called BRG1 in order to act as a pioneer transcription factor. BRG1 is an enzyme that is able to move and remove (remodel) nucleosomes attached to DNA, suggesting that normal transcription factor binding requires this activity. The next challenge is to investigate whether BRG1, or similar enzymes, are also needed by other pioneer transcription factors that are required for normal gene activity and cell identity. This will be important because many enzymes that remodel nucleosomes are disrupted in human diseases like cancer where cells lose their normal identity.

even when nucleosomes are present (*Zaret and Carroll, 2011*). This is exemplified by the forkhead box A1 (FoxA1) transcription factor, which binds to and de-compacts nucleosomal DNA in vitro (*Cirillo et al., 2002*) and in vivo (*Holmqvist et al., 2005*; *Lupien et al., 2008*; *Iwafuchi-Doi et al., 2016*). Based on this type of activity, factors like FoxA1 have been called 'pioneer' transcription factors as they appear to play a primary role in recognizing and shaping how new gene regulatory elements are established in previously inaccessible chromatin (*Magnani et al., 2011*; *Zaret and Carroll, 2011*). Once bound to their target sites, pioneer transcription factors appear to create accessible chromatin (*Raposo et al., 2015*; *Schulz et al., 2015*) that supports the recruitment of non-pioneer transcription factors and the formation of functional gene regulatory elements (*Theodorou et al., 2013*; *Wapinski et al., 2013*; *Foo et al., 2014*; *Xu et al., 2014*; *Schulz et al., 2015*). These unique features of pioneer transcription factors allow them to function as master regulators that underpin developmental transitions, cellular reprogramming, and responses to cellular signalling events.

The capacity of pioneer transcription factors to destabilise or reposition nucleosomes to facilitate chromatin accessibility at their target sites appears to be an essential feature of their activity. Initially, it was proposed that pioneer factors might achieve this by interacting specifically with nucleosomes to alter chromatin structure. For example, FoxA1 and FoxO1 structurally resemble histone H1, which has been proposed to lead to displacement of H1 and destabilisation of neighbouring nucleosomes (*Cirillo et al., 2002*; *Hatta and Cirillo, 2007*). Alternatively, it has been proposed that ATP-dependent chromatin remodellers may be required to assist pioneer transcription factors in establishing accessible chromatin (*Hu et al., 2011*; *Marathe et al., 2013*; *Ceballos-Chávez et al., 2015*; *Swinstead et al., 2016*). Nevertheless, for most pioneer transcription factors there remains limited understanding of the mechanisms by which they translate their binding within inaccessible and nucleosome-occluded chromatin into effects on chromatin structure that support additional transcription factor binding and gene regulation.

One paradigm that has been used to study pioneer transcription factor function is the cellular reprogramming of somatic cells into induced pluripotent stem cells (iPSC) by the Yamanaka transcription factors (OCT4, SOX2, KLF4, and c-MYC) (*Takahashi and Yamanaka, 2006*). With the

exception of c-MYC, these transcription factors are proposed to act as pioneers during cellular reprogramming and can bind their targets sites even when they are occupied by nucleosomes (*Soufi et al., 2012*, *2015*; *Chen et al., 2016*). It is thought that this pioneering activity may then pave the way for nucleosome displacement and further transcription factor binding that establishes functional regulatory elements (*You et al., 2011*; *Shakya et al., 2015*; *Simandi et al., 2016*). While it is clear that these transcription factors have the capacity to engage with previously inaccessible regions of chromatin during iPSC reprogramming, it is unknown whether additional mechanisms are required to transition these initial engagement events into functionally mature transcription factor binding as part of the pluripotency-associated regulatory network.

To begin addressing this fundamental gap in our understanding, we have used mouse embryonic stem cells, which exist in an established pluripotent state, as a model system to dissect how the core pluripotency transcription factor OCT4 engages with target sites inside cells. In doing so we discover that OCT4 occupies sites that would otherwise be inaccessible and is required to shape the occupancy of additional pluripotency transcription factors. OCT4 achieves this by recruiting the chromatin remodelling factor BRG1 to support not only its own binding but also to stabilize further transcription factor binding events required to support pluripotency-associated gene regulation. This reveals that although OCT4 can engage with its target sites in inaccessible chromatin, the recruitment of a chromatin remodelling enzyme is a fundamental step in pluripotency transcription factor binding and gene expression. Intriguingly, OCT4-bound regulatory sites that require BRG1 are activated more slowly in response to cellular reprogramming and later during early development. Together these observations reveal that the capacity of OCT4 to cooperate with a chromatin remodeller is a key feature of its pioneering activity, and this is required to mature transcription factor binding and create functional gene regulatory elements.

## Results

### The pioneer factor OCT4 binds distal regulatory sites in pluripotent cells that would otherwise be inaccessible

The atomic structure of OCT4 and its DNA binding activity in vitro have been characterized in detail (*Klemm et al., 1994*; *Esch et al., 2013*), yet the mechanisms which support how OCT4 functions as a pioneer transcription factor in vivo remain poorly defined. Our current understanding is based largely on overexpression studies during iPSC reprogramming where OCT4 binds a large number of its motifs in inaccessible regions of chromatin (*Soufi et al., 2012*, *2015*; *Chen et al., 2016*). However, whether these binding events represent functionally relevant pioneering interactions that precede the creation of accessible chromatin and downstream target gene expression remains largely unknown. We therefore set out to examine how OCT4 engages with and functions at its binding sites in a more physiologically relevant situation in mouse embryonic stems cells (ESCs), where OCT4 plays an essential role in shaping distal regulatory element function and controlling the pluripotent transcriptome. In order to achieve this, we first set out to identify a *bona fide* set of OCT4 binding events in mouse ESCs using chromatin immunoprecipitation coupled with massively parallel sequencing (ChIP-seq). We applied this approach to a conditional mouse ESC line where addition of a small molecule (doxycycline) leads to loss of OCT4 expression (*Niwa et al., 2000*). It is known that prolonged removal of OCT4 in ESCs results in loss of pluripotency and cellular differentiation (*Niwa et al., 2000*; *Adachi et al., 2013*). We therefore identified an acute treatment condition where following 24 hr of doxycycline treatment cells lacked appreciable OCT4 protein (*Figure 1A*) but retained normal ESC morphology, were alkaline phosphatase positive, and expressed wild type levels of the pluripotency transcription factors SOX2 and NANOG (*Figure 1A,B*). Analysis of our OCT4 ChIP-seq identified 15,920 high-confidence OCT4 binding sites that were lost following doxycycline treatment (*Figure 1C*) and were highly enriched for known OCT4 binding motifs (*Figure 1— figure supplement 1A,B*). The majority of these binding events (75%) correlated with a histone modification signature usually associated with distal regulatory elements (high H3K4me1/low H3K4me3), while only a small subset (6.8%) corresponded to sites with a promoter associated histone modification signature (high H3K4me3/low H3K4me1) (*Figure 1D*; *Figure 1—figure supplement 1C,D*). These observations are consistent with previous reports indicating that OCT4 binds extensively to distal as opposed to promoter proximal regulatory regions in the genome (*Chen et al., 2008*;

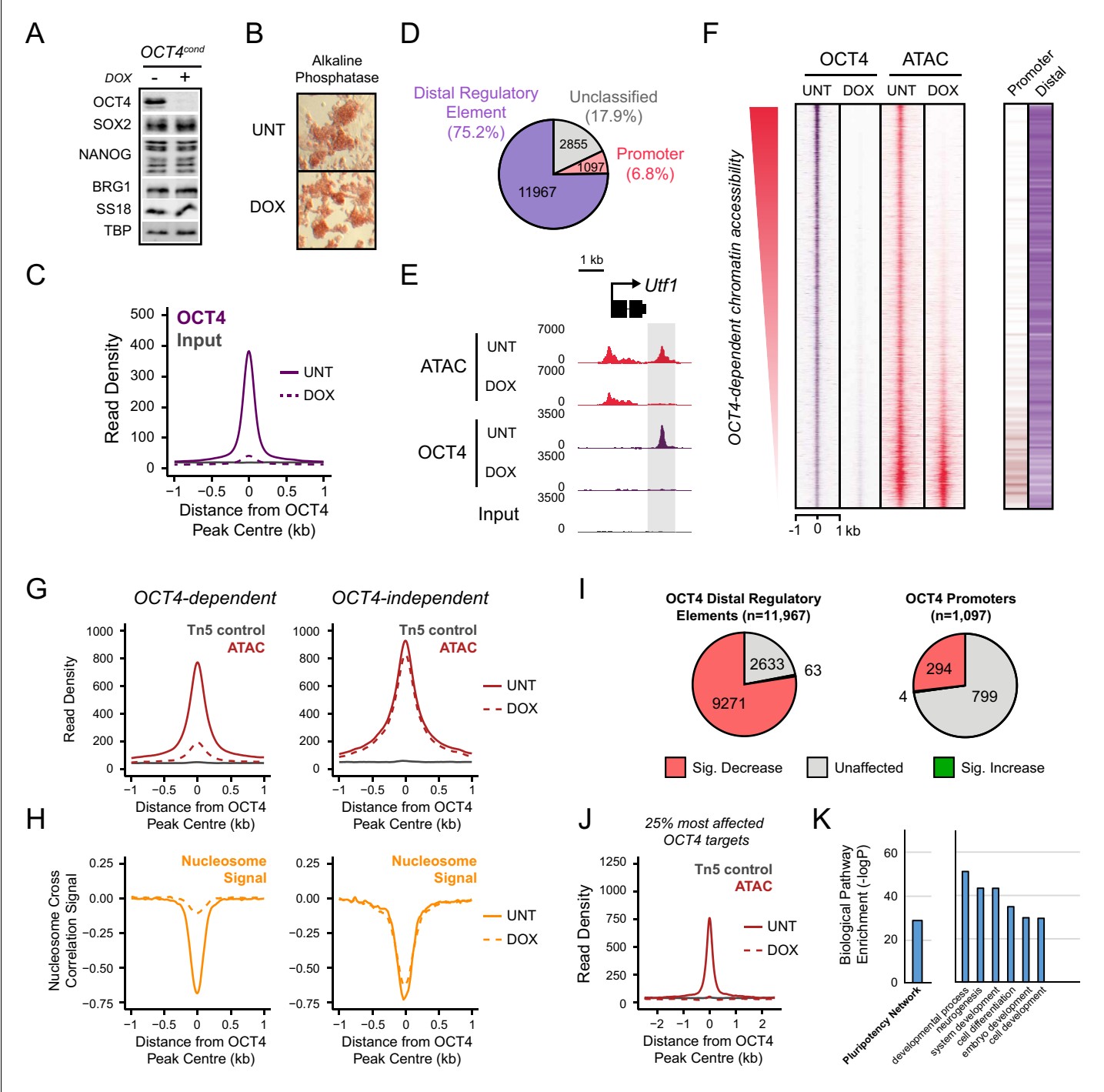

**Figure 1.** OCT4 binds distal regulatory sites in mouse embryonic stems cells to shape chromatin accessibility. (**A**) Western blot analysis of *OCT4cond* (ZHBTC4) mouse ESCs before (UNT) and after 24 hr treatment with doxycycline (DOX). (**B**) Alkaline phosphatase staining of *OCT4cond* mouse ESCs before (UNT) and after 24 hr DOX treatment. (**C**) A metaplot of OCT4 ChIP-seq signal in *OCT4cond* ESCs before (UNT) and after 24 hr DOX treatment at OCT4 peaks (*n* = 15920). (**D**) Annotation of OCT4 peaks as promoters or distal regulatory elements using the relative enrichment of promoter-associated H3K4me3 or distal regulatory element-associated H3K4me1. (**E**) A genomic snapshot of ATAC-seq and OCT4 ChIP-seq signal in *OCT4cond* ESCs before (UNT) and after 24 hr DOX treatment at the *Utf1* locus. The downstream OCT4-bound regulatory element is highlighted in the grey box. (**F**) A heatmap illustrating OCT4 targets (*n* = 15920) ranked by their loss of chromatin accessibility (ATAC-seq) after 24 hr DOX treatment of *OCT4cond* ESCs. Normalised read densities for ATAC-seq and OCT4 ChIP-seq are presented, with a heatmap indicating their annotation as either promoters or distal regulatory elements (right). (**G**) A metaplot of *OCT4cond* ATAC-seq signal before (UNT) and after 24 hr DOX treatment at OCT4 binding sites with significant reductions in ATAC-seq signal (OCT4-dependent; *n* = 11557) and those without significant changes (OCT4-independent; *n* = 4362). Tn5

*Figure 1 continued on next page*

*Figure 1 continued*

control represents transposition of purified genomic DNA to control for potential sequence bias. (H) As in (G), profiling the changes in nucleosome occupancy before (UNT) and after (DOX) OCT4 depletion. Nucleosome signal was generated using the NucleoATAC package. (I) Piecharts identifying the proportion of OCT4-bound distal regulatory elements (left) or OCT4-bound promoters (right) that display significant changes in chromatin accessibility as measured by ATAC-seq. Changes were deemed to be significant with *FDR* < 0.05 and a fold change greater than 1.5-fold. (J) A metaplot depicting the *OCT4^cond* ATAC-seq signal before (UNT) and after (DOX) treatment at the 25% of OCT4 peaks with the greatest changes in ATAC-seq signal following OCT4 depletion. (K) Gene ontology analysis for genes closest to OCT4 target sites depicted in (J). This reveals an enrichment for the pluripotency expression network (left) and biological processes associated with developmental gene regulation (right).

The following figure supplements are available for figure 1:

**Figure supplement 1.** Annotation and characterisation of OCT4 binding sites in *OCT4^cond* ESCs.

**Figure supplement 2.** Changes in chromatin accessibility at OCT4-bound sites following depletion of OCT4 in ESCs.

**Figure supplement 3.** Chromatin accessibility profiling at OCT4 binding sites in somatic cell lines or tissues.

*Göke et al., 2011*). The identification of *bona fide* OCT4 target sites, and the maintenance of stem cell features under these treatment conditions, provided us with an opportunity to examine in more detail where and how OCT4 normally engages with the ESC genome, and to ask how this is related to underlying chromatin accessibility and transcription factor co-occupancy.

During somatic cell reprogramming, exogenous OCT4 is proposed to function as a pioneer transcription factor that can bind to its sequence motifs in inaccessible regions of chromatin. However, it remains unclear whether binding to inaccessible chromatin is also a feature of normal OCT4 binding in mouse ESCs. To address this important question we used the assay for transposase-accessible chromatin with massively parallel sequencing (ATAC-seq) which provides a genome-wide measure of chromatin accessibility (*Buenrostro et al., 2013*) and examined ATAC-seq signal in wild type and OCT4-depleted cells. Although OCT4-bound sites were highly accessible in wild type cells, when we examined ATAC-seq signal in the OCT4-depleted ESCs, 72% of OCT4 targets showed significant reductions in chromatin accessibility (*Figure 1E,F,G* and *Figure 1—figure supplement 2*) and increases in nucleosome occupancy (*Figure 1H*). These observations are in agreement with previous studies describing a role for OCT4 in maintaining nucleosome-depleted regions and/or chromatin accessibility at individual loci in pluripotent cells (*You et al., 2011*; *Shakya et al., 2015*) or genome-wide (*Chen et al., 2014*; *Lu et al., 2016*). Importantly, OCT4-bound distal regulatory elements appeared to be most significantly affected, while OCT4-bound promoters experienced few significant reductions in accessibility (*Figure 1I*). Consistent with a pioneering-like role for OCT4 in shaping chromatin structure, many OCT4-bound regulatory elements were completely inaccessible in the OCT4-depleted ESCs (*Figure 1F,J*) and lacked any detectable chromatin accessibility in cells and tissues lacking OCT4 expression (*Figure 1—figure supplement 3*). Importantly, OCT4 binding sites that displayed reduced accessibility following OCT4 removal were often in close proximity with genes implicated in the pluripotency regulatory network (*Figure 1K*), suggesting that these OCT4 binding events may be implicated with the maintenance of pluripotency-associated gene expression. These observations therefore establish that the majority of OCT4 binding events in pluripotent stem cells occur at sites that would otherwise be inaccessible and occupied by nucleosomes, indicating that OCT4 is required to maintain accessible chromatin at its target sites not only during cellular reprogramming but also in the established pluripotent state.

## OCT4 supports transcription factor binding at distal regulatory elements to regulate pluripotency-associated genes

A defining feature of pioneer transcription factors is their capacity to support additional transcription factor occupancy, potentially through alteration of local chromatin structure. Given that removal of OCT4 had widespread effects on chromatin accessibility at its binding sites, we wondered whether its absence would result in defects in the binding of other pluripotency-associated transcription factors. To address this possibility, we used ChIP-seq to examine the binding of SOX2 and NANOG in the presence or absence of OCT4 (*Figure 2*). Interestingly, we observed major reductions in SOX2

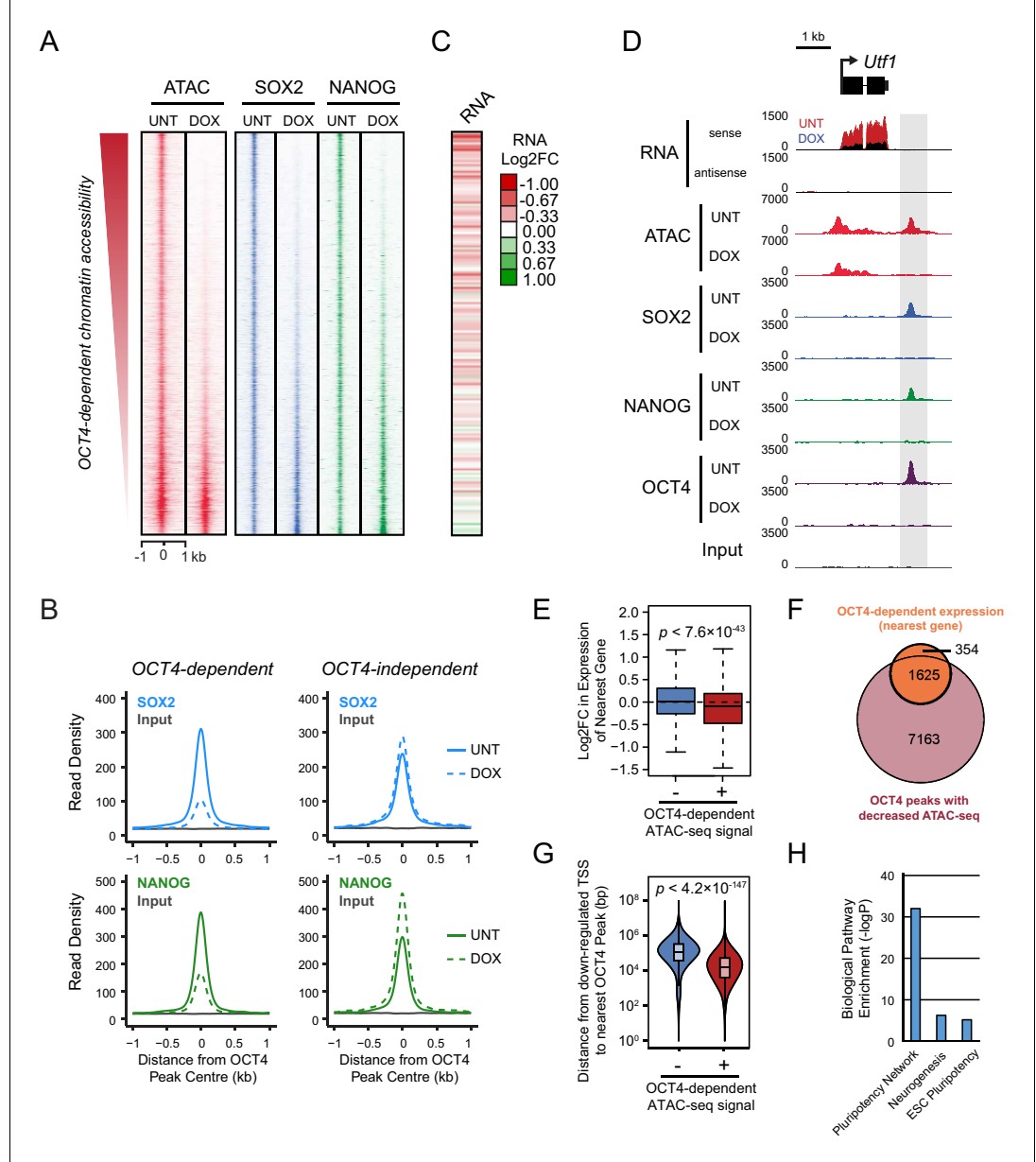

**Figure 2.** Loss of OCT4 leads to reduced pluripotency-associated transcription factor binding and expression of nearby genes. (A) A heatmap illustrating SOX2 and NANOG ChIP-seq at OCT4 targets (n = 15920) ranked by their loss of chromatin accessibility (ATAC-seq) after 24 hr DOX treatment of $OCT4^{cond}$ ESCs, as in *Figure 1F*. (B) A metaplot of $OCT4^{cond}$ SOX2 and NANOG ChIP-seq signal before (UNT) and after 24 hr DOX treatment at OCT4 binding sites with significant reductions in ATAC-seq signal (OCT4-dependent; n = 11557) and those without significant changes (OCT4-independent; n = 4362). (C) A heatmap of the log2 fold change in RNA-seq signal of the closest gene to OCT4 target sites depicted in (A), following 24 hr DOX treatment of the $OCT4^{cond}$ ESCs. (D) A genomic snapshot of *Utf1*, which is decreased in expression following loss of OCT4. The distal OCT4 target site is highlighted by a grey box. (E) Comparison of log2 fold change (log2FC) in RNA-seq signal for genes neighbouring OCT4 target sites that rely on OCT4 for ATAC-seq signal (OCT4-dependent; n = 11557) or those that do not (OCT4-independent; n = 4362). (F) A Venn diagram comparing the OCT4 binding sites for which the nearest gene has significantly reduced expression after OCT4 ablation (orange; n = 1979) and the OCT4 targets with significant reductions in chromatin accessibility (red; n = 8788). Only OCT4 target sites for which the nearest gene has sufficient RNA-seq coverage are included. (G) A violin plot comparing the distance from OCT4 binding sites with OCT4-dependent (n = 11557) or OCT4-independent (n = 4362) chromatin accessibility to nearest TSS with significant reductions in RNA-seq following 24 hr DOX treatment of the $OCT4^{cond}$ ESCs (n = 1430). (H) Gene ontology analysis for genes down-regulated after loss of OCT4 (n = 1430) reveals enrichment of the pluripotency transcriptional network.

The following figure supplement is available for figure 2:

*Figure 2 continued on next page*

*Figure 2 continued*

**Figure supplement 1.** Loss of SOX2 and NANOG is highly correlated with reductions in chromatin accessibility at OCT4-SOX2-NANOG targets.

and NANOG binding at the OCT4 target sites that experienced the largest reductions in chromatin accessibility (*Figure 2A,B* and *Figure 2—figure supplement 1A,B*). Importantly, the reductions in SOX2 and NANOG binding correlated extremely well with the reductions in chromatin accessibility at OCT4-bound sites (*Figure 2—figure supplement 1C*). Conversely, the smaller subset of OCT4-bound sites that retained chromatin accessibility following OCT4 depletion retained SOX2 or NANOG occupancy (*Figure 2B*). This suggests that OCT4 plays a central role in supporting combinatorial binding and chromatin accessibility at the majority of OCT4 binding sites.

To understand whether this loss of combinatorial transcription factor binding was relevant to gene regulation, we carried out nuclear RNA-seq in wild type and OCT4-depleted cells. Loss of OCT4-dependent chromatin accessibility and transcription factor binding was broadly associated with the down-regulation of nearby genes (*Figure 2C–E*), such as the pluripotency-associated *Utf1* gene (*Figure 2D*). When we examined this relationship across the genome, reductions in the expression of genes near OCT4-bound distal sites was highly coincident (82%) with reductions in chromatin accessibility (*Figure 2F*). Interestingly, however, reductions in chromatin accessibility at an OCT4-bound site did not always lead to alterations in the expression of neighbouring gene, suggesting that some bound sites may not function in gene regulation or that they may have regulatory capacity that extends beyond the nearest gene. Nevertheless, genes that were down-regulated in OCT4-depleted ESCs (*n* = 1430; OCT4-dependent gene expression) tended to be significantly closer to OCT4 binding sites that displayed reduced chromatin accessibility (median distance 16.2 kb) than OCT4 binding sites that retained chromatin accessibility in the absence of OCT4 (median distance 125.4 kb) (*Figure 2G*). These sites were also characterized by reductions in SOX2 and NANOG binding (*Figure 2A,B*) and their associated genes were enriched for the pluripotency-associated transcriptional network (*Figure 2H*). Together, these observations reveal that OCT4 binding plays a primary and widespread role in shaping combinatorial binding of transcription factors at otherwise inaccessible regulatory sites in ESCs, and this activity underpins pluripotency-associated gene expression.

## The chromatin remodelling enzyme BRG1 is enriched at sites where OCT4 is responsible for chromatin accessibility

Several modalities have been proposed to explain how pioneer factors facilitate chromatin accessibility at otherwise inaccessible regions of the genome. For example, pioneer transcription factors may bind their target motifs and evict or exclude nucleosomes through steric mechanisms (*Cirillo et al., 2002*; *Hatta and Cirillo, 2007*; *Voss and Hager, 2014*) or they may exploit the activity of chromatin remodelling enzymes (*Marathe et al., 2013*; *Ceballos-Chávez et al., 2015*; *Swinstead et al., 2016*). However, the relative contribution of such mechanisms to pioneer transcription factor function remains poorly understood and unaddressed for OCT4. What is clear from our analysis in ESCs is that the majority of OCT4-bound sites exist in an inaccessible chromatin state in its absence, and that OCT4 plays a primary role in supporting SOX2 and NANOG binding at these sites. To examine whether this pioneering-like activity could potentially rely on the activity of ATP-dependent chromatin remodelling enzymes, we took advantage of the extensive ChIP-seq analysis of chromatin remodelling enzymes that exists for mouse ESCs (*Wang et al., 2014*; *de Dieuleveult et al., 2016*), and simply examined across the complete complement of accessible sites in the mouse ESC genome whether there was a relationship between binding of any these enzymes and the dependency on OCT4 for chromatin accessibility. Remarkably, of the nine individual chromatin remodelling complexes examined, only BRG1 (SMARCA4), a catalytic ATPase subunit of the vertebrate SWI/SNF chromatin remodelling complex (also referred to as BRG1-associated factor (BAF) remodelling complexes), showed an appreciable correlation between occupancy and the reliance on OCT4 for chromatin accessibility (*Figure 3A*). Indeed, BRG1 was significantly enriched at regulatory elements that relied on OCT4 for chromatin accessibility (OCT4-dependent ATAC peaks) and regulatory elements bound by OCT4 (*Figure 3B–D*). Furthermore, OCT4 and BRG1 occupancy was highly

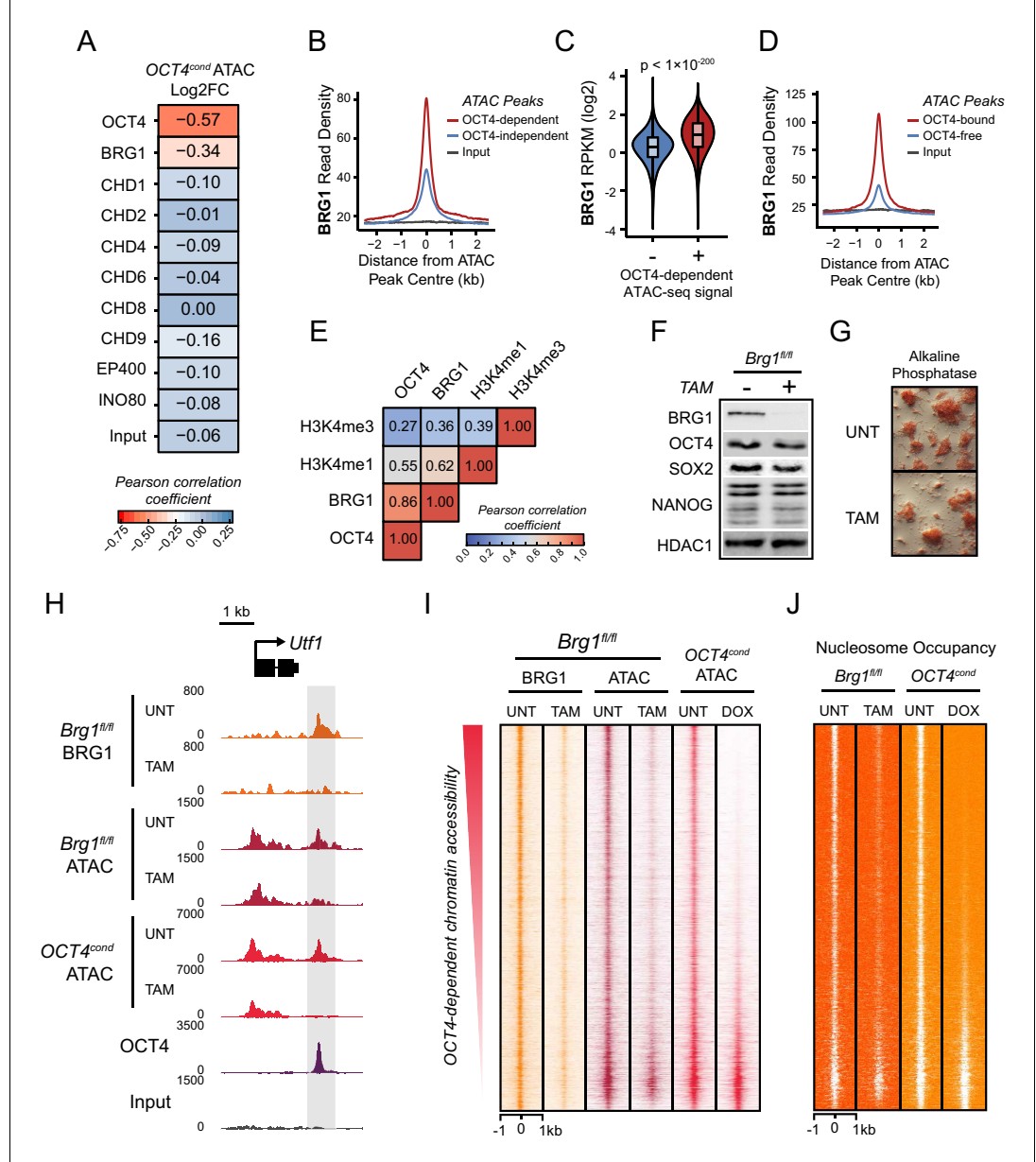

**Figure 3.** The chromatin remodelling enzyme BRG1 is required to create accessible chromatin at OCT4 target sites. (A) A Pearson correlation matrix comparing log2 fold change in ATAC-seq signal in OCT4-depleted cells with wild type ESC ChIP-seq signal for nine chromatin remodellers at wild type ATAC hypersensitive peaks (*n* = 76,642). (B) A metaplot of BRG1 ChIP-seq signal at ATAC hypersensitive peaks with (OCT4-dependent) or without (OCT4-independent) significant reduction in ATAC-seq signal following removal of OCT4. (C) A violin plot quantifying and comparing BRG1 ChIP-seq reads per kilobase per million (RPKM) at OCT4-dependent or OCT4-independent ATAC-seq peaks depicted in (B). (D) A metaplot of BRG1 ChIP-seq signal at OCT4-bound or OCT4-free ATAC-seq peaks. (E) Genome-wide correlation of OCT4, BRG1, H3K4me1 and H3K4me3 in 2 kb windows reveals a high degree of co-localization between OCT4 and BRG1. (F) Western blot analysis for the indicated proteins in *Brg1^fl/fl* mouse ESCs before (UNT) and after 72 hr tamoxifen (TAM) treatment. (G) Alkaline phosphatase staining of *Brg1^fl/fl* ESCs before (UNT) and after 72 hr TAM treatment. (H) A genomic snapshot of BRG1 ChIP-seq and ATAC-seq in *Brg1^fl/fl* ESCs before (UNT) and after 72 hr TAM treatment at the distal OCT4 target site downstream of *Utf1* (highlighted in grey). The *OCT4^cond* ATAC-seq is included for comparison and reveals a co-dependency on OCT4 and BRG1 for normal chromatin accessibility. (I) A heat map of BRG1 ChIP-seq and ATAC-seq at OCT4 target sites (*n* = 15920) in *Brg1^fl/fl* ESCs before (UNT) and after 72 hr TAM treatment. Sites are ranked by loss of ATAC-seq signal following removal of OCT4, as in **Figure 1F**, and the *OCT4^cond* ATAC-seq is included for comparison. (J) As in (I), changes in nucleosome occupancy before (UNT) and after (TAM) BRG1 depletion are plotted based on nucleosome signal derived from the NucleoATAC package.

The following figure supplement is available for figure 3:

*Figure 3 continued on next page*

*Figure 3 continued*

**Figure supplement 1.** OCT4 target sites require BRG1 to maintain chromatin accessibility.

correlated throughout the mouse ESC genome (*Figure 3E* and *Ho et al. (2009)*, *Kidder et al. (2009)*, *de Dieuleveult et al., 2016*), supporting the possibility that there may be a functional link between OCT4 binding, BRG1, and chromatin accessibility.

## BRG1 is required to create accessible chromatin at OCT4 target sites in ESCs

BRG1, like OCT4, is essential for maintaining ESC pluripotency (*Ho et al., 2009*; *Kidder et al., 2009*; *Zhang et al., 2014*), supporting early embryonic development (*Bultman et al., 2006*) and improving the efficiency of iPSC reprogramming (*Singhal et al., 2010*). However, the defined molecular mechanisms by which BRG1 and the BAF complex contribute to maintenance of the pluripotent ESC state remain unclear. Based on our observation that BRG1 was enriched at sites that rely on OCT4 binding for chromatin accessibility (*Figure 3A–C*), we reasoned that OCT4 may require BRG1 to shape chromatin accessibility and gene regulatory capacity of OCT4-bound sites in maintaining the pluripotent state. To address this possibility we performed ATAC-seq on a conditional mouse ESC line in which BRG1 expression can be conditionally ablated following addition of tamoxifen (*Brg1*$^{fl/fl}$) (*Ho et al., 2009*). Tamoxifen treatment resulted in near complete loss of BRG1 by 72 hr, but importantly, BRG1-depleted cells retained normal ESC morphology, were alkaline phosphatase positive, and expressed wild type levels of OCT4, SOX2 and NANOG as described previously (*Ho et al., 2011*) (*Figure 3F–G*). When we examined ATAC-seq signal at several OCT4 target sites in the BRG1-depleted ESCs, we observed substantial reductions in chromatin accessibility compared to the untreated control ESCs, and these effects were similar to the reductions observed in the OCT4-depleted ESCs (*Figure 3H*; *Figure 3—figure supplement 1A*). When we extended this analysis to all OCT4-bound sites, it was clear that chromatin accessibility was reduced (*Figure 3I*; *Figure 3—figure supplement 1B*) and nucleosome occupancy increased following BRG1 depletion at the majority of OCT4 target sites (*Figure 3J*; *Figure 3—figure supplement 1C*). We then used a stringent threshold (fold change >1.5 and *FDR* < 0.05) to identify sites that experienced significant reductions in chromatin accessibility following BRG1 removal. This revealed that 45% of OCT4-bound sites showed significantly reduced ATAC-seq signal and these sites were also highly dependent on OCT4 for their accessibility (*Figure 3—figure supplement 1D*). However, we suspected that the use of a threshold to identify affected sites might underestimate the extent of BRG1's contribution. Indeed, when we applied an unbiased clustering approach to examine the relationship between removal of OCT4 and BRG1 in shaping accessibility it was clear that the majority (76%) of OCT4-bound sites that rely on OCT4 for their accessibility were also dependent on BRG1 for their accessibility (*Figure 3—figure supplement 1E*). Importantly, the loss of either OCT4 or BRG1 appeared to result in similar, although not identical, reductions in chromatin accessibility across nearly all OCT4 peaks (*Figure 3I*; *Figure 3—figure supplement 1F*), revealing that both of these factors are required to maintain chromatin accessibility at these loci in mouse ESCs. Together, these observations suggest that BRG1 plays a widespread role in shaping chromatin accessibility at OCT4 target sites in ESCs and raises the interesting possibility that the pioneering-like activity of OCT4 may rely on BRG1.

## OCT4 establishes chromatin accessibility by recruiting BRG1 to chromatin

Our ATAC-seq experiments revealed an essential role for BRG1 in regulating chromatin accessibility at OCT4 target sites in ESCs. One could envisage several possible mechanisms by which BRG1 could cooperate with OCT4 to achieve this. For example, BRG1 could be actively recruited by OCT4 to target sites in order to create accessible chromatin, or it could alternatively function to broadly remodel the genome and in doing so indirectly support access of OCT4 to its target sites. Given that large scale proteomic studies have previously indicated that BRG1 may interact physically with OCT4 (*Pardo et al., 2010*; *van den Berg et al., 2010*; *Ding et al., 2012*), and BRG1 is enriched at OCT4-

bound sites throughout the genome (*Figure 3*), we favoured the possibility that OCT4 recruits BRG1 and the associated BAF complex to OCT4 target sites in the genome in order to remodel chromatin and promote chromatin accessibility. In fitting with this possibility, OCT4, BRG1 and the additional BAF subunit, SS18, are enriched within the nucleosome-depleted region at OCT4 target sites (*Figure 4A*). To address whether OCT4 was responsible for recruiting BRG1 and BAF complexes to the distal regulatory sites it binds, we performed ChIP-seq for BRG1 and SS18 in wild type and OCT4-depleted ESCs. Although BRG1 and SS18 protein levels were unaffected following removal of OCT4 (*Figure 1A*), we observed a dramatic reduction in BRG1 and SS18 binding at OCT4 target sites in the absence of OCT4 (*Figure 4B,C*). This loss was specific to OCT4-bound regulatory elements as regulatory elements lacking OCT4 were unaffected (*Figure 4D*). Together, these

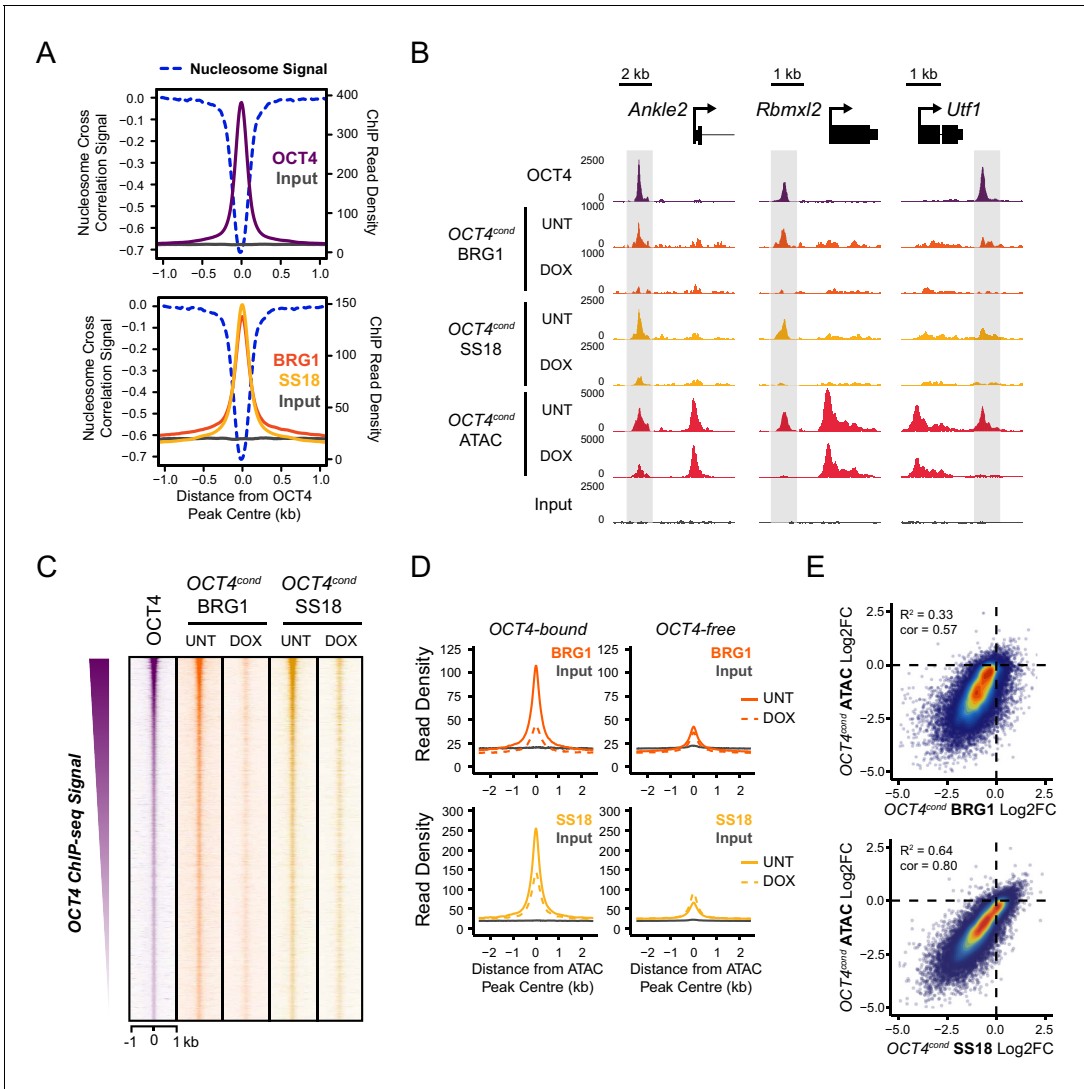

**Figure 4.** OCT4 is required for normal BRG1 chromatin occupancy at OCT4-bound regulatory elements. (A) A high resolution metaplot illustrating nucleosome occupancy, OCT4, BRG1 and BRG1-associated factor SS18 at OCT4 peaks (n = 15920; 10 bp windows) demonstrates that OCT4 and BRG1/BAF signal co-localises to the nucleosome-depleted region at OCT4 peaks. (B) Genomic snapshots of BRG1 and SS18 ChIP-seq in *OCT4^cond* before (UNT) and after 24 hr doxycycline (DOX) treatment reveals loss of BRG1 and SS18 occupancy at distal OCT4 targets (highlighted in grey) following OCT4 removal. (C) A heatmap of OCT4 peaks (*n* = 15920) illustrating enrichment of BRG1 and SS18 at OCT4 target sites in wild type cells and subsequent loss of BRG1 following removal of OCT4. (D) A metaplot of BRG1 and SS18 ChIP-seq signal at ATAC hypersensitive sites with (OCT4-bound) or without (OCT4-free) OCT4 before and after deletion of OCT4. (E) A scatterplot comparing the changes in BRG1 and SS18 occupancy with the changes in chromatin accessibility (ATAC-seq) at OCT4 peaks after deletion of OCT4. $R^2$ represents linear regression score, and *cor* reflects Pearson correlation coefficient.

observations suggested that OCT4 shapes chromatin accessibility at its target sites through the recruitment of BRG1/BAF complex. Indeed, when we compared reductions in chromatin accessibility with reductions in BRG1 and SS18 at OCT4 binding sites, there was a good correlation between the loss of these factors and loss of ATAC-seq signal in OCT4-depleted ESCs (*Figure 4E*). Interestingly, our observation that OCT4 recruits the BRG1/BAF complex to its target sites in ESCs is in agreement with recent observations that BRG1 is recruited to pluripotency-associated enhancers coincident with the formation of accessible chromatin during iPSC reprogramming (*Chronis et al., 2017*). Together these analyses indicate that OCT4 plays a central role in recruiting BRG1-containing BAF complexes to pluripotency-associated gene regulatory elements and that this is important for creating accessible chromatin at these sites.

## BRG1 supports transcription factor binding at distal gene regulatory elements

We have identified a widespread and important role for the chromatin remodeller BRG1 in creating accessibility at gene regulatory elements bound by OCT4 in ESCs. Given that OCT4 recruits BRG1 to shape chromatin accessibility (*Figure 4*), we wondered whether BRG1 was also important to sustain the engagement and function of OCT4, its binding partner SOX2 and the additional pluripotency associated transcription factor NANOG. To examine this interesting possibility, we carried out ChIP-seq for OCT4, SOX2 and NANOG in the BRG1 conditional cells before and after tamoxifen treatment (*Figure 5*). Importantly, wild type OCT4 ChIP-seq signal was highly similar in both *Brg1$^{fl/fl}$* ESCs and *OCT4$^{cond}$* ESCs (*Figure 5—figure supplement 1*). This allowed us to examine the contribution of BRG1 to transcription factor occupancy at *bona fide* OCT4 binding sites (*Figure 1*), focusing on OCT4-bound distal regulatory elements as these were the sites most dependent on OCT4 and BRG1 for normal chromatin accessibility. Strikingly, we observed significant reductions in OCT4 binding at the majority (60%; *FDR* < 0.05 and fold change >1.5 fold) of distal OCT4 targets following BRG1 removal (*Figure 5A–C*), indicating that BRG1 contributes not only to creating accessibility at OCT4 target sites but is also required to sustain normal OCT4 binding in mouse ESCs. Similarly, SOX2 and NANOG binding was also significantly reduced following BRG1 removal in ESCs, although the total number of sites significantly affected was less than for OCT4 (16.7% and 22.9% of distal OCT4 target sites; *FDR* < 0.05 and fold change >1.5 fold) (*Figure 5A–C*). Importantly, even when we examined the most extremely affected OCT4 target sites (BRG1-dependent), loss of BRG1 did not result in complete loss of transcription factor binding (*Figure 5D*), yet resulted in an inability of the cells to maintain the nucleosome-depleted state (*Figure 5—figure supplement 2*). This suggests that OCT4, SOX2 and NANOG may still be able to recognise their sequence motifs, but are unable to engage in normal and robust binding without the cooperation of BRG1/BAF remodelling complexes. Ultimately, this is consistent with a pioneering activity of OCT4 being required to initially sample and engage with inaccessible chromatin, potentially through the recognition of partial DNA motifs (*Soufi et al., 2015*). However, OCT4-dependent BRG1/BAF recruitment appears to subsequently be required to functionally mature these previously inaccessible sites such that they can now effectively support robust binding of OCT4 and additional pluripotency-associated transcription factors.

## Gene expression defects in the absence of BRG1 are linked to altered transcription factor binding

In the absence of BRG1, the stable binding of pluripotency transcription factors OCT4, SOX2 and NANOG was disrupted at distal regulatory elements, suggesting that this may affect expression of the pluripotency-associated transcriptional network. Somewhat surprisingly, previous work has not supported a clear correlation between loss of BRG1 and reduced activity of OCT4 target genes (*Ho et al., 2009*, *2011*; *Zhang et al., 2014*; *Hainer et al., 2015*), despite our observations that BRG1 plays an important role in transcription factor binding at many distal regulatory elements in close proximity to pluripotency-associated genes. However, we also observed that the effect on OCT4, SOX2 and NANOG binding following BRG1 removal varied in magnitude between individual sites, with some sites actually displaying increases in transcription factor binding (*Figure 5A–D*). We therefore reasoned that transcriptional effects following loss of BRG1 might not precisely correlate

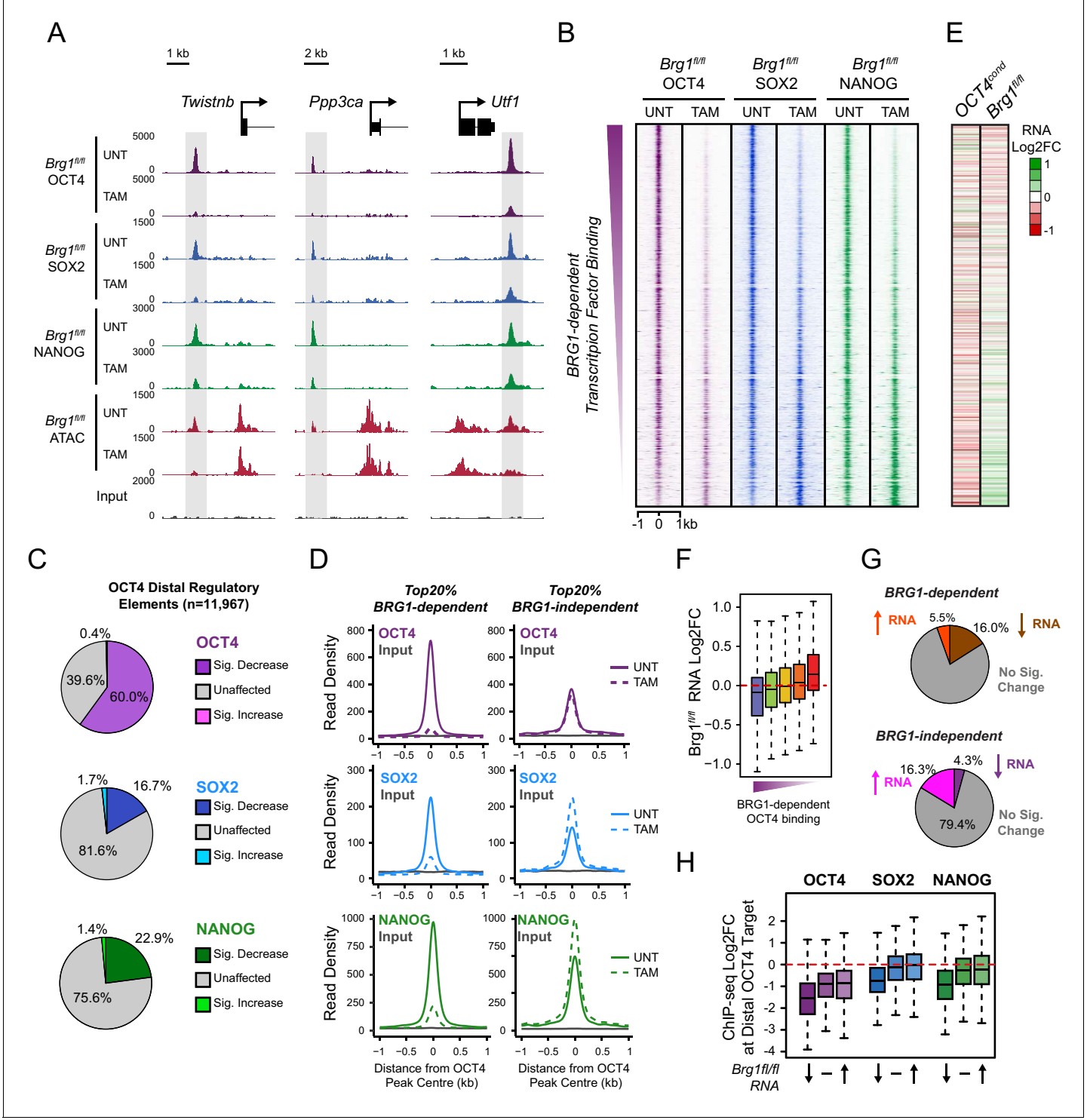

**Figure 5.** BRG1 supports pluripotency-associated transcription factor binding to functionally mature distal gene regulatory elements. (**A**) Genomic snapshots illustrating OCT4, SOX2 and NANOG ChIP-seq signal in *Brg1*^fl/fl^ ESCs before (UNT) and after tamoxifen (TAM) treatment for 72 hr. Three examples of distal OCT4 targets are depicted, with affected sites highlighted in grey. (**B**) A heatmap analysis of OCT4, SOX2 and NANOG ChIP-seq signal at distal OCT4 targets (*n* = 11967) ranked by their relative change in OCT4 ChIP-seq after deletion of BRG1. (**C**) Piecharts depicting the significant changes in OCT4, SOX2 and NANOG ChIP-seq signal at distal OCT4 target sites, identified using the DiffBind package. Changes were deemed significant with *FDR* < 0.05 and a change greater than 1.5-fold. (**D**) Metaplot analysis of OCT4, SOX2 and NANOG binding at distal OCT4 target sites that are the most (20% most affected; BRG1-dependent) and the least (20% least affected; BRG1-independent) dependent on BRG1 for normal OCT4 binding. (**E**) A heatmap illustrating the changes in gene expression (log2 fold change in RNA-seq) for genes neighbouring distal OCT4

*Figure 5 continued on next page*

*Figure 5 continued*

targets. Genes are sorted according to the change in OCT4 occupancy at neighbouring distal OCT4 target sites following BRG1 removal, as in (B). Gene expression changes for OCT4-depleted (*OCT4$^{cond}$*) or BRG1-depleted (*Brg1$^{fl/fl}$*) ESCs relative to wildtype ESCs are shown. (F) Quantitation of log2 fold change (log2FC) in *Brg1$^{fl/fl}$* RNA-seq signal for genes neighbouring distal OCT4 target sites grouped into quintiles based on the change in OCT4 binding at neighbouring distal OCT4 target sites following removal of BRG1, as in (B). (G) Comparison of significant gene expression changes for genes neighbouring distal OCT4 target sites that show the largest (20% most affected; BRG1-dependent) and least (20% least affected; BRG1-independent) reductions in OCT4 ChIP-seq signal following BRG1 removal. Changes were deemed significant with *FDR* < 0.05 and a change greater than 1.5-fold, using DESeq2. (H) Quantitation of log2 fold change (log2FC) of OCT4, SOX2 and NANOG ChIP-seq signal at distal OCT4 targets in proximity to OCT4-dependent genes with decreased (↓; n = 468), unchanged (-; n = 639), or increased (↑; n = 816) RNA-seq signal after deletion of BRG1 (as identified in *Figure 5—figure supplement 3B*).

The following figure supplements are available for figure 5:

**Figure supplement 1.** Comparison of wild type OCT4 ChIP-seq signal between *OCT4$^{cond}$* and *Brg1$^{fl/fl}$* ESCs.

**Figure supplement 2.** Nucleosome occupancy changes in *Brg1$^{fl/fl}$* ESCs at distal OCT4 target sites.

**Figure supplement 3.** Transcriptional regulation of OCT4-dependent target genes by BRG1.

**Figure supplement 4.** Changes in gene expression following depletion of BRG1 in *Brg1$^{fl/fl}$* ESCs are associated with altered transcription factor binding.

with the expression changes following OCT4 removal but may instead be related to alterations in OCT4, SOX2 and NANOG binding at individual regulatory sites.

To examine this possibility, we carried out nuclear RNA-seq in wild type and BRG1-depleted cells, and compared gene expression changes to those observed following conditional removal of OCT4 (*Figure 5E* and *Figure 5—figure supplement 3*). As described earlier (*Figure 2*), genes in close proximity to OCT4 binding sites tended to be down-regulated following loss of OCT4 (*Figure 5E*). When we simply overlapped the genes that showed significant reductions in gene expression following OCT4 removal (OCT4-dependent genes; *FDR* < 0.05 and fold change >1.5) with genes that significantly changed in expression following BRG1 removal, the overlap was low (*Figure 5—figure supplement 3A*). This was in agreement with previous work that did not identify a clear correlation between gene expression changes following OCT4 and BRG1 removal (*Ho et al., 2009*, *2011*; *Zhang et al., 2014*; *Hainer et al., 2015*). However, when we examined this relationship in more detail using unbiased clustering of the OCT4-dependent genes based on their expression changes after removal of BRG1, we identified three separate types of response; genes that showed reduced expression (28.1%), genes whose expression was unchanged (35.2%), and genes that had increased gene expression (36.7%) (*Figure 5—figure supplement 3B,C*). Given that individual distal regulatory elements vary in their requirement for BRG1 in OCT4, SOX2 and NANOG binding (*Figure 5A–D*), we examined whether these changes in gene expression corresponded to changes in transcription factor binding at nearby distal regulatory elements. Importantly, this analysis revealed that the expression of genes associated with sites that rely on BRG1 for OCT4 binding tended to be reduced following deletion of BRG1 (*Figure 5E–G*, *Figure 5—figure supplement 4*). In contrast, genes associated with unchanged or increased OCT4, SOX2 and NANOG binding tended to show increases in expression (*Figure 5E–G*, *Figure 5—figure supplement 4*). Furthermore, distal OCT4 binding sites in close proximity to OCT4/BRG1 dependent genes experienced the largest reductions in transcription factor binding following removal of BRG1 (*Figure 5H*). Therefore, our new genome-wide analysis suggests that altered transcription factor binding is a major determinant of the gene expression changes in BRG1-depleted ESCs, with the direction and magnitude of expression change being dictated by effects on transcription factor binding at nearby distal regulatory elements. Importantly, these observations support a model where BRG1 is required to stabilize and mature pioneer binding events by OCT4 at inaccessible distal regulatory elements and to transition these sites into functionally active regulatory elements that control the transcription of nearby genes. In contrast, some distal regulatory elements rely less on BRG1 for transcription factor binding and exhibit less pronounced reductions, or even increases, in the expression of their associated genes. Together our

new analyses explain why previous studies had failed to identify a simple relationship between gene expression changes following OCT4 and BRG1 removal and demonstrate that reduced expression of a subset of OCT4 target genes results from the inability of OCT4 and other transcription factors to bind their target sites following BRG1 removal.

## BRG1-dependency reveals distinct modes of OCT4 function at distal regulatory elements during reprogramming and development

Through studying OCT4 function in iPSC reprogramming it has been proposed that there may be distinct phases of OCT4 binding which are influenced by pre-existing chromatin states in somatic cells (*Soufi et al., 2012*; *Buganim et al., 2013*; *Chen et al., 2016*). For example, OCT4 occupies a subset of sites during the early stages of reprogramming which are modified by histone H3 lysine 4 dimethylation or histone H3 lysine 27 acetylation and associated with more accessible chromatin. This is followed by OCT4 binding at sites lacking pre-existing chromatin modifications that are thought to become accessible during the later stages of reprogramming when pluripotency is established. This suggests that OCT4 binding may occur via different mechanisms to support regulatory element function during reprogramming and development.

In light of our observation in ESCs that some OCT4 binding does not require BRG1, we wanted to examine whether the requirement for BRG1 at these sites reflected the dynamics of gene expression during cellular reprogramming and early development. We therefore analysed gene expression during the reprogramming of mouse fibroblasts into iPSCs via OCT4/SOX2/KLF4/MYC from two independent studies (*Chen et al., 2016*; *Cieply et al., 2016*). Specifically, we examined genes that required OCT4 for their expression in ESCs (OCT4-dependent) and separated these into genes associated with BRG1-independent or BRG1-dependent OCT4 binding at neighbouring distal regulatory elements in ESCs (see *Figure 5*). We then compared the timing of their expression during the iPSC reprogramming process. This revealed that genes associated with BRG1-independent OCT4 binding were activated early during reprogramming and BRG1-dependent sites later (*Figure 6A*). This suggests that OCT4 expression in somatic cells leads to a more immediate transcriptional response from genes associated with BRG1-independent OCT4 binding, presumably due to pre-existing chromatin accessibility at these loci (*Chen et al., 2016*). In contrast, genes that require BRG1 for OCT4 binding were expressed later during reprogramming, perhaps because OCT4 requires BRG1 at these sites to remodel chromatin and mature the function of these regulatory elements. In fitting with this possibility, BRG1-dependent OCT4 binding sites were also associated with genes expressed at later developmental stages during early mouse development (*Figure 6B*), suggesting that chromatin state in the early embryo may also shape how these OCT4 target sites are used during development. To explore this possibility, we examined the activation of BRG1-independent and BRG1-dependent OCT4 targets in the early mouse embryo using chromatin accessibility as a proxy for OCT4 binding and regulatory activity (*Lu et al., 2016*; *Wu et al., 2016*). Interestingly, this demonstrated that BRG1-dependent OCT4 target sites became accessible at later stages of embryonic development (*Figure 6C*, *Figure 6—figure supplement 1*). This suggests that developmental transitions may require both OCT4-dependent chromatin binding and BRG1-dependent remodelling activities to overcome the activation barrier set by chromatin. In contrast, the pre-existing chromatin state at a subset of OCT4 target sites (BRG1-independent) may allow them to be activated more rapidly. Importantly, these observations suggest that chromatin structure likely plays an important role in regulating how OCT4 engages with and functions in the genome not only in mouse ESCs but also during reprogramming and development.

## Discussion

Although the capacity of pioneer transcription factors to bind their sequence motifs in chromatinized DNA has been studied in detail, the mechanisms that support how individual pioneer transcription factors function to create accessible chromatin and shape regulatory element function remain poorly defined. Through studying the pioneer transcription factor OCT4, we establish that OCT4 binds to sites in mouse ESCs that would otherwise be inaccessible to shape chromatin accessibility and transcription factor binding (*Figures 1* and *2*). This suggests that the pioneering activities of OCT4 are required not only during reprogramming, but also in the established pluripotent state. Interestingly, chromatin accessibility formed at OCT4 binding sites relies on the chromatin remodelling factor

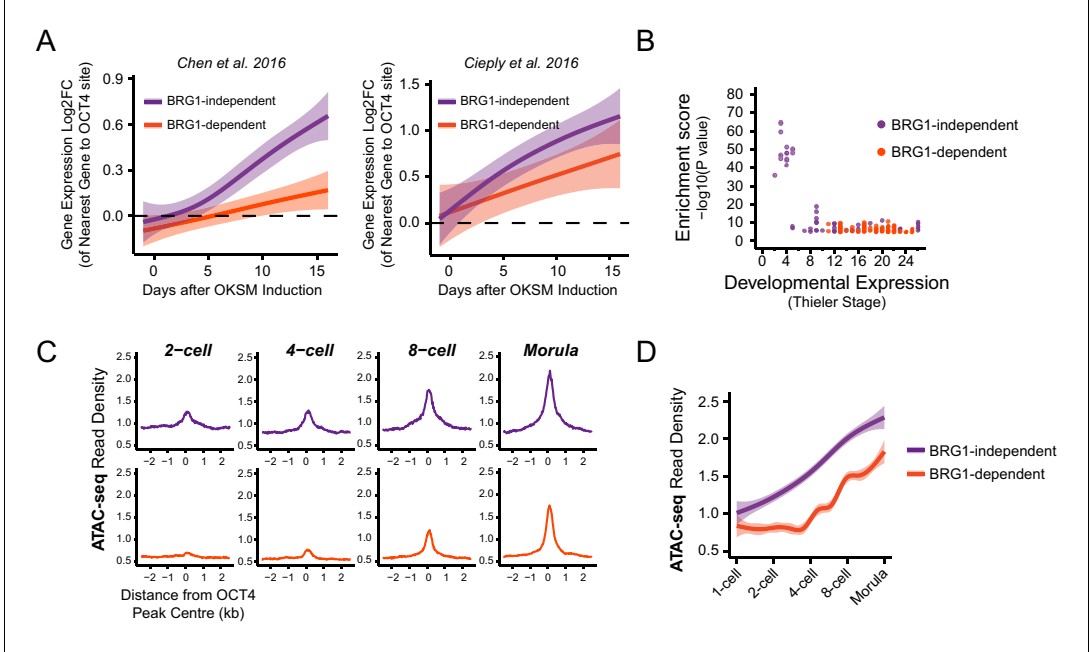

**Figure 6.** BRG1-dependency reveals distinct modes of OCT4 function of distal regulatory elements during reprogramming and development. (**A**) A time course of gene expression changes (log2 fold change) during OKSM-mediated reprogramming of mouse embryonic fibroblasts to iPSCs (*Chen et al., 2016*; *Cieply et al., 2016*). The expression changes of OCT4-dependent genes neighbouring the 20% of distal OCT4 targets that were most dependent upon BRG1 for normal OCT4 binding (BRG1-dependent) or the 20% of distal OCT4 targets that were least dependent (BRG1-independent) following BRG1 removal were quantified and visualised as a smoothed trendline ±95% confidence interval. (**B**) Genomic Regions Enrichment of Annotations Tool (GREAT) annotation of BRG1-dependent and BRG1-independent distal OCT4 targets. Plotted are the enrichment (-log10(P Value)) against the MGI Expression profile, or Thieler development stage, for genes neighbouring distal OCT4 targets. BRG1-independent sites are strongly enriched for gene expression profiles consistent with very early embryonic stages. (**C**) Metaplot profiles of ATAC-seq signal during early mouse embryonic development (*Wu et al., 2016*) at BRG1-independent (upper) and BRG1-dependent (lower) distal OCT4 target sites. BRG1-independent sites gain accessibility earlier than BRG1-dependent sites. (**D**) A quantitation of ATAC-seq read density at distal OCT4 targets depicted in (**C**) during early embryonic development plotted as a smoothed trendline ±95% confidence interval.

The following figure supplement is available for figure 6:

**Figure supplement 1.** BRG1-dependent and BRG1-independent OCT4 binding at distal regulatory elements is associated with distinct developmental timing of chromatin accessibility.

BRG1 (*Figure 3*) which is recruited to these sites by OCT4 (*Figure 4*). The occupancy of BRG1 is then required to support efficient OCT4, SOX2 and NANOG binding and normal expression of the pluripotency-associated transcriptome (*Figure 5*). Importantly, this reliance on BRG1 reflects OCT4 binding dynamics during cellular reprogramming and early mouse development (*Figure 6*). Together these observations reveal a distinct requirement for the chromatin remodelling factor BRG1 in shaping the activity of the pioneer transcription factor OCT4 and regulating the pluripotency network in embryonic stem cells.

Pioneer transcription factors play very defined roles in shaping gene expression in response to cellular reprogramming (*Soufi et al., 2012*; *Raposo et al., 2015*; *Boller et al., 2016*; *Chen et al., 2016*), developmental stimuli, and environmental cues (*Hu et al., 2011*; *Wu et al., 2011*; *Ballaré et al., 2013*; *Schulz et al., 2015*). While considerable effort has focused on characterizing the chromatin binding activity of pioneer transcription factors, both in vitro and in vivo, the mechanisms by which individual pioneer transcription factors shape chromatin structure and support further transcription factor engagement in vivo have remained poorly understood. Our examination of OCT4 binding now provides a potential rationalization for how this pioneer transcription factor functions to form gene regulatory elements in mouse ESCs. A requirement of this process is OCT4's ability to engage with sequences in inaccessible chromatin presumably through its capacity to bind

nucleosomal DNA (*Figure 1*; *Soufi et al., 2012*, *2015*), suggesting that OCT4 may be able to dynamically sample its target sites throughout the genome. However, cooperation with BRG1 appears to be required for more stable OCT4 binding (*Figure 5*). This could be achieved through the chromatin remodelling activity of BRG1 creating transiently accessible DNA that facilitates further OCT4 binding and ultimately more stable cooperative binding with other transcription factors like SOX2 and NANOG. This general model is consistent with previous experiments that showed enhanced OCT4 binding during OCT4-SOX2-KLF4 reprogramming when BRG1 was included in the reprogramming cocktail (*Singhal et al., 2010*) and is in agreement with our observation that BRG1 is required for normal chromatin accessibility and OCT4 binding at otherwise inaccessible and inactive regulatory elements.

The capacity of OCT4 to recognise its target motifs in inaccessible chromatin may allow it to dynamically sample its complement of binding sites in the genome. However, like many transcription factors, OCT4 only binds stably to a subset of these sites. Our genome-wide analyses indicate that BRG1 and transcription factor co-binding events appear to be essential in stabilizing OCT4 binding and the functional maturation of OCT4-dependent regulatory elements in ESCs. Based on these observations, one would predict that co-binding transcription factors might play a central role in shaping where OCT4 stably engages with the genome in different cell types. In fitting with this possibility, when ESCs are transitioned into epiblast-like cells (EpiLCs), OCT4 stably associates with a new set of previously inaccessible sites in a manner that appears to rely on co-binding of the EpiLC-specific transcription factor OTX2 (*Buecker et al., 2014*; *Yang et al., 2014*). In the context of these and other observations, it is tempting to speculate that OCT4, through its association with BRG1, is exploited as a dynamic pioneering and chromatin remodelling module by distinct co-binding transcription factors to support the formation of cell type-specific regulatory elements during developmental transitions and reprogramming. In agreement with these general ideas, OCT4 appears to be the only Yamanaka transcription factor that cannot be substituted for in iPSC reprogramming experiments, suggesting its pioneering activity is fundamental in establishing and maintaining the pluripotency network in concert with SOX2, KLF4 and c-Myc (*Nakagawa et al., 2008*; *Sterneckert et al., 2012*), perhaps due to its cooperation with BRG1 (*Singhal et al., 2010*; *Esch et al., 2013*). This central requirement in reprogramming is also consistent with our observations that OCT4 plays an essential and BRG1-dependent role in shaping the binding of other pluripotency-associated factors at thousands of distal regulatory elements in ESCs.

One of the central features of pioneer transcription factors is their ability to alter local chromatin structure, and this is closely linked to their capacity to support further transcription factor binding and enable the functional maturation of distal regulatory elements. Yet, in most instances the defined molecular mechanisms that support these activities have remained undetermined. While some pioneer factors, such as FoxA1, can stably bind their target sites and directly alter nucleosomal structure through steric disruption of histone:DNA contacts (*Cirillo et al., 2002*; *Hatta and Cirillo, 2007*; *Iwafuchi-Doi et al., 2016*), the extent to which other pioneer factors might exploit such a mechanism has yet to be fully addressed. In the case of OCT4, our observations establish that its capacity to support transcription factor binding and chromatin accessibility requires the chromatin remodeller BRG1. This suggests that unlike FoxA1, OCT4 does not have an intrinsic chromatin opening activity, but requires cooperation with BRG1 to achieve this. Indeed, our work on OCT4 is generally consistent with and supported by recent work studying the pioneer factor GATA3 which appears to have similar dependency on BRG1 in creating accessible chromatin (*Takaku et al., 2016*). Furthermore, chromatin remodelling activity has been widely implicated in shaping nucleosome positioning and chromatin accessibility at target sites bound by other pioneer transcription factors (*Hu et al., 2011*; *Sanalkumar et al., 2014*; *Ceballos-Chávez et al., 2015*; *Hainer and Fazzio, 2015*). Together, these observations suggest that many pioneer transcription factors may rely on chromatin remodelling as a key step in supporting transcription factor binding and/or co-binding events in a manner broadly consistent with the previously proposed assisted loading model for transcription factor binding (*Voss and Hager, 2014*; *Swinstead et al., 2016*). In the context of this model, the dynamic binding, release and re-binding of transcription factors would help to maintain and stabilise transcription factor binding events, but critically, chromatin remodellers appear to be important in many cases to achieve this (reviewed recently in *Swinstead et al., 2016*). Clearly more work is required to understand the extent to which pioneer transcription factors use chromatin remodellers to support their pioneering activities. However, in the case of the developmental pioneer transcription factors OCT4

and GATA3, chromatin remodellers act as key component necessary for the formation and function of gene regulatory elements. These emerging observations suggest that chromatin remodelling could function as a central feature of pioneer transcription factor activity during the formation and maintenance of cell type-specific transcriptional programs during development and reprogramming.

## Materials and methods

### Cell culture and lines

Mouse embryonic stem cell (ESCs) containing a doxycycline-sensitive OCT4 transgene (ZHBTC4; referred to here as $OCT4^{cond}$ [*Niwa et al., 2000*]) were grown on gelatin-coated plates in DMEM (Gibco, Carlsbad, CA) supplemented with 15% FBS, 10 ng/mL leukemia-inhibitory factor, penicillin/ streptomycin, beta-mercaptoethanol, L-glutamine and non-essential amino-acids. $OCT4^{cond}$ cells were treated with 1 µg/mL doxycycline for 24 hr to ablate OCT4 expression, which was verified by Western blotting. $Brg1^{fl/fl}$ ESCs were previously described (*Ho et al., 2011*) and maintained in DMEM KnockOut supplemented with 10% FBS and 5% KnockOut Serum Replacement (Life Technologies, Carlsbad, CA), plus additional factors described for $OCT4^{cond}$ ESCs above. $Brg1^{fl/fl}$ ESCs were treated with 4-hydroxytamoxifen for 72 hr to ablate BRG1 protein levels, which was verified by Western blotting. Cell lines were routinely tested and confirmed to be mycoplasma-free. Alkaline phosphatase staining was performed by incubating cells with freshly prepared AP buffer (0.4 mg/mL naphthol phosphate N-5000 (Sigma, St Louis, MO), 1 mg/mL Fast Violet B Salt F-161 (Sigma), 100 mM Tris-HCl (pH 9.0), 100 mM NaCl, 5 mM $MgCl_2$).

### ATAC-seq sample preparation and sequencing

Chromatin accessibility was assayed using an adaptation of the assay for transposase accessible-chromatin (ATAC)-seq (*Buenrostro et al., 2013*). Briefly, $5 \times 10^6$ cells were harvested, washed with PBS and nuclei were isolated in 1 mL HS Lysis buffer (50 mM KCl, 10 mM $MgSO_4.7H_20$, 5 mM HEPES, 0.05% NP40 [IGEPAL CA630]), 1 mM PMSF, 3 mM DTT) for 1 min at room temperature. Nuclei were centrifuged at $1000 \times g$ for 5 min at 4°C, followed by a total of three washes with ice-cold RSB buffer (10 mM NaCl, 10 mM Tris (pH 7.4), 3 mM $MgCl_2$), to remove as much cytoplasmic and mitochondrial material as possible. Nuclei were then counted, and $5 \times 10^4$ nuclei were resuspended in Tn5 reaction buffer (10 mM TAPS, 5 mM $MgCl_2$, 10% dimethylformamide) and 2 µl of Tn5 transposase (25 µM) made in house according to the previously described protocol (*Picelli et al., 2014*). Nuclei were then incubated for 30 min at 37°C, before isolation and purification of tagmented DNA using QiaQuick MinElute columns (Qiagen, Germany). To control for sequence bias of the Tn5 transposase, a Tn5 digestion control was performed by tagmenting ESC genomic DNA with Tn5 for 30 min at 55°C. ATAC-seq libraries were prepared by PCR amplification using custom made Illumina barcodes previously described (*Buenrostro et al., 2013*) and the NEBNext High-Fidelity 2X PCR Master Mix (NEB, Ipswich, MA) with 8–10 cycles. Libraries were purified with two rounds of Agencourt AMPure XP bead cleanup (1.5X beads:sample; Beckman Coulter, Brea, CA), followed by quantification by qPCR using SensiMix SYBR (Bioline, UK) and KAPA Library Quantification DNA standards (KAPA Biosystems, Wilmington, MA). ATAC-seq libraries were sequenced on Illumina NextSeq500 using 80 bp paired-end reads in biological triplicate.

### Chromatin immunoprecipitation and ChIP-seq library preparation

Chromatin immunoprecipitation (ChIP) was performed as described previously (*Farcas et al., 2012*), with minor modifications. Cells were fixed for 1 hr in 2 mM DSG and 12.5 min in 1% formaldehyde. Reactions were quenched by the addition of glycine to a final concentration of 125 µM. After cell lysis and chromatin extraction, chromatin was sonicated using a BioRuptor sonicator (Diagenode, Belgium), followed by centrifugation at $16,000 \times g$ for 20 min at 4°C, and used fresh or stored at −80°C. Chromatin was quantified by denaturing chromatin 1:10 in 0.1M NaOH and measuring DNA concentration by NanoDrop. 150 µg chromatin/IP was diluted ten-fold in ChIP dilution buffer (1% Triton-X100, 1 mM EDTA, 20 mM Tris-HCl (pH 8), 150 mM NaCl) prior to pre-clearing with prepared protein A magnetic Dynabeads (Invitrogen, Carlsbad, CA) which had been blocked for 1 hr with 0.2 mg/mL BSA and 50 µg/mL yeast tRNA. Chromatin samples were then incubated overnight with relevant antibodies at 4°C with rotation. Antibodies used for ChIP experiments were

anti-OCT4A (Cell Signaling Technology (CST, Danvers, MA), #5677), anti-SOX2 (CST, #23064), anti-Nanog (CST, #8822), anti-SS18 (CST, #21792) and anti-BRG1 (abcam (UK), ab110641). Antibody-bound chromatin was isolated on protein A magnetic Dynabeads. ChIP washes were performed with low salt buffer (0.1% SDS, 1% Triton, 2 mM EDTA, 20 mM Tris-HCl (pH 8.1), 150 mM NaCl), high salt buffer (0.1% SDS, 1% Triton, 2 mM EDTA, 20 mM Tris-HCl (pH 8.1), 500 mM NaCl), LiCl buffer (0.25M LiCl, 1% NP40, 1% sodium deoxycholate, 1 mM EDTA, 10 mM Tris-HCl (pH 8.1)) and TE buffer (x2 washes) (10 mM Tris-HCl (pH 8.0), 1 mM EDTA). To prepare ChIP-seq material, ChIP DNA was eluted using 1% SDS and 100 mM $NaHCO_3$, and cross-links reversed at 65°C in the presence of 200 mM NaCl. Samples were then treated with RNase and proteinase K before being purified with ChIP DNA Clean and Concentrator kit (Zymo, Irvine, CA). ChIP-seq libraries were prepared using the NEBNext Ultra DNA Library Prep Kit with NEBNext Dual Indices, and sequenced as 38 bp paired-end reads on Illumina NextSeq500 platform. All ChIP-seq experiments were carried out in biological triplicate.

## Nuclear RNA-seq sample generation and sequencing

To isolate nuclear RNA, cells were subjected to nuclei isolation described for ATAC-seq. Nuclei were then resuspended in TriZOL reagent (ThermoScientific, Waltham, MA) and RNA was extracted according to the manufacturer's protocol. Nuclear RNA was treated with the TURBO DNA-free Kit (ThermoScientific) and depleted for rRNA using the NEBNext rRNA Depletion kit and protocol (NEB). RNA-seq libraries were prepared using the NEBNext Ultra Directional RNA-seq kit (NEB) and libraries were sequenced on the Illumina NextSeq500 with 80 bp paired-end reads in biological triplicate.

## Massively parallel sequencing, data processing and normalisation

For ATAC-seq and ChIP-seq, paired-end reads were aligned to the mouse mm10 genome using bowtie2 (*Langmead and Salzberg, 2012*) with the '–no-mixed' and '–no-discordant' options. Non-uniquely mapping reads and reads mapping to a custom 'blacklist' of artificially high regions of the genome were discarded. For RNA-seq, reads were initially aligned using bowtie2 against the rRNA genomic sequence (GenBank: BK000964.3) to filter out rRNA fragments, prior to alignment against the mm10 genome using the STAR RNA-seq aligner (*Dobin et al., 2013*). To improve mapping of nascent, intronic sequences, reads which failed to map using STAR were aligned against the genome using bowtie2. PCR duplicates were removed using SAMtools (*Li et al., 2009*). Biological replicates were randomly downsampled to contain the same number of reads for each individual replicate, and merged to create a representative genome track using DANPOS2 (*Chen et al., 2013*) which was visualised using the UCSC Genome Browser. Peakcalling analyses were performed using the DAN-POS2 dpeak function on untreated and treated samples in biological triplicate with matched input. Merged ATAC-seq datasets were used to extract signal corresponding to nucleosome occupancy information with NucleoATAC (*Schep et al., 2015*) using a cross correlation model for all regulatory elements (ATAC hypersensitive sites) in each cell line.

## Differential binding and gene expression analysis

Significant changes in ATAC-seq or ChIP-seq datasets were identified using the DiffBind package (*Stark and Brown, 2011*), while for RNA-seq DESeq2 was used with a custom-built, non-redundant mm10 gene set (*Love et al., 2014*). Briefly, mm10 refGene genes were filtered on size (>200 bp), gene body and TSS mappability, unique TSS and TTS, in order to remove poorly mappable and highly similar transcripts. For both DiffBind and DESeq2, a *FDR* < 0.05 and a fold change >1.5 fold was deemed to be a significant change. For distal OCT4 intervals, gene expression changes for the nearest TSS were considered. *K*-means clustering to identify OCT4 and BRG1 co-dependency of chromatin accessibility or gene expression was performed using the kmeans function in R. Clusters with similar trends (i.e. decrease, no change, or increase) were then grouped together for subsequent analysis. Changes in ATAC-seq or ChIP-seq were visualised using heatmaps or metaplots produced using HOMER2 (*Heinz et al., 2010*), with heatmaps made using Java TreeView (*Saldanha, 2004*). Log2 fold change values were visualised using R (v 3.2.1), with scatterplots coloured by density using stat_density2d. Regression and correlation analyses were also performed in R using standard linear models and Pearson correlation respectively. Log2 fold change or reads per

kilobase per million (RPKM) values were compared between different classes of transcription factor binding sites either by visualising *gam* smoothed trendlines with 95% confidence intervals or using the Wilcoxon signed-rank test.

## Functional annotation of transcription factor binding sites

OCT4 peaks were identified in the $OCT4^{cond}$ biological triplicate OCT4 ChIP-seq data using the DANPOS2 dpeak function and only peaks with decreased OCT4 ChIP-seq signal were taken for further analysis ($n$ = 15920). OCT4 motif enrichment analysis was performed using the MEME suite (*Bailey et al., 2009*). Briefly, Analysis of Motif Enrichment (AME) for canonical motifs was performed in parallel to de novo motif identification with Discriminative Regular Expression Motif Elicitation (DREME) using the central 200 bp of OCT4 peaks. Putative de novo motifs were further subjected to CentriMo analysis to identify motifs that were enriched for the centre of OCT4 peaks. OCT4 peaks were annotated as putative promoters or putative distal regulatory elements in a manner similar to that described previously (*Hay et al., 2016*), using the relative and absolute coverage of H3K4me3 (a promoter-associated modification; *Yue et al., 2014*) and H3K4me1 (associated with distal regulatory elements; *Whyte et al., 2012*). Transcription factor peaks with different characteristics were analysed using the GREAT package (*McLean et al., 2010*), in particular to extract information regarding developmental expression timing from MGI Gene eXpression Database. HOMER2 was used to identify the nearest transcription start sites (TSS) of genes and to perform gene ontology (GO) analysis for differentially regulated genes. Comparison of different remodelling complexes was performed by calculating RPKM for chromatin remodeller ChIP-seq in mouse ESCs across all ATAC peaks ($n$ = 76,642) and determining the Pearson correlation with the log2 fold change of $OCT4^{cond}$ ATAC-seq. Genome-wide correlation of BRG1, OCT4, H3K4me3 and H3K4me1 was generated using the bamCorrelate function of deepTools (*Ramírez et al., 2014*).

## Accession numbers

ATAC-seq, ChIP-seq and RNA-seq data from the present study are available for download at GSE87822. Previously published datasets used for analysis include mouse ESC H3K4me1 (GSE27844; *Whyte et al., 2012*) and H3K4me3 ChIP-seq (GSE49847; *Yue et al., 2014*), ENCODE DNase-seq (GSE37074; *Yue et al., 2014*), ESC chromatin remodeller ChIP-seq (GSE49137, GSE64825; *Wang et al., 2014*; *de Dieuleveult et al., 2016*), iPSC expression data (GSE67462, GSE70022; *Chen et al., 2016*; *Cieply et al., 2016*), early mouse embryo ATAC-seq (GSE66581; *Wu et al., 2016*) and DNase-seq (GSE76642; *Lu et al., 2016*).

## Acknowledgements

Work in the Klose lab is supported by the Wellcome Trust, the Lister Institute of Preventive Medicine, and Exeter College University of Oxford, EMBO, and the European Research Council. We would like to thank Tatyana Nesterova and Neil Brockdorff for the kind gift of the ZHBTC4/$OCT4^{cond}$ mouse embryonic stem cells, and Gerald Crabtree for generously providing us with the $Brg1^{fl/fl}$ mouse embryonic stem cells and discussing results prior to publication. We would also like to thank Brian Hendrich, Neil Blackledge, Vincenzo di Cerbo and Guifeng Wei for critical reading of the manuscript, and Emilia Dimitrova and Anne Turberfield for helpful discussion and comments.

## Additional information

### Funding

| Funder | Grant reference number | Author |
| --- | --- | --- |
| Wellcome | 098024/Z/11/Z | Robert J Klose |
| European Research Council | 681440 | Robert J Klose |
| University of Oxford | Exeter College, University of Oxford, Monsanto Senior Research Fellowship | Robert J Klose |
| Lister Institute of Preventive | | Robert J Klose |

| University of Oxford | Exeter College, University of Oxford, Monsanto Senior Research Fellowship | Robert J Klose |
|---|---|---|
| Medicine | | |

The funders had no role in study design, data collection and interpretation, or the decision to submit the work for publication.

## Author contributions

HWK, Conceptualization, Data curation, Software, Formal analysis, Funding acquisition, Validation, Investigation, Visualization, Methodology, Writing—original draft, Writing—review and editing; RJK, Supervision, Funding acquisition, Writing—original draft, Project administration, Writing—review and editing

## Author ORCIDs

Hamish W King, http://orcid.org/0000-0001-5972-8926
Robert J Klose, http://orcid.org/0000-0002-8726-7888

# Additional files

## Major datasets

The following dataset was generated:

| Author(s) | Year | Dataset title | Dataset URL | Database, license, and accessibility information |
|---|---|---|---|---|
| King HW, Klose RJ | 2017 | The pioneer factor OCT4 requires the chromatin remodeller BRG1 to support gene regulatory element function in mouse embryonic stem cells | https://www.ncbi.nlm.nih.gov/geo/query/acc.cgi?acc=GSE87822 | Publicly available at the NCBI Gene Expression Omnibus (accession no: GSE87822) |

The following previously published datasets were used:

| Author(s) | Year | Dataset title | Dataset URL | Database, license, and accessibility information |
|---|---|---|---|---|
| ENCODE | 2014 | A comparative encyclopedia of DNA elements in the mouse genome | https://www.ncbi.nlm.nih.gov/geo/query/acc.cgi?acc=GSE49847 | Publicly available at the NCBI Gene Expression Omnibus (accession no: GSE49847) |
| Young R | 2012 | Enhancer Decommissioning by LSD1 During Embryonic Stem Cell Differentiation | https://www.ncbi.nlm.nih.gov/geo/query/acc.cgi?acc=GSE27844 | Publicly available at the NCBI Gene Expression Omnibus (accession no: GSE27844) |
| ENCODE | 2012 | DNaseI Hypersensitivity by Digital DNaseI from ENCODE/University of Washington | https://www.ncbi.nlm.nih.gov/geo/query/acc.cgi?acc=gse37074 | Publicly available at the NCBI Gene Expression Omnibus (accession no: GSE37074) |
| Wang L, Hu G, Du Y | 2014 | INO80 complex in the core regulatory network governing ESC self-renewal [ChIP-Seq] | https://www.ncbi.nlm.nih.gov/geo/query/acc.cgi?acc=GSE49137 | Publicly available at the NCBI Gene Expression Omnibus (accession no: GSE49137) |
| Hmitou I, de Dieu-leveult M, Depaux A, Chantalat S, Yen K, Pugh BF, Gérard M | 2016 | Genome-wide distribution and function of ATP-dependent chromatin remodelers in embryonic stem cells | https://www.ncbi.nlm.nih.gov/geo/query/acc.cgi?acc=GSE64825 | Publicly available at the NCBI Gene Expression Omnibus (accession no: GSE64825) |
| Xing Y, Carstens R | 2016 | Multiphasic and dynamic changes | https://www.ncbi.nlm. | Publicly available at |

| | | | | |
|---|---|---|---|---|
| | | | in alternative splicing during induction of pluripotency are coordinated by numerous RNA binding proteins [iPS] | nih.gov/geo/query/acc. cgi?acc=GSE70022 | the NCBI Gene Expression Omnibus (accession no: GSE70022) |
| Li M | | 2015 | Expression data from OSKM-mediated 2nd reprogramming cells and the corresponding iPS cell line | https://www.ncbi.nlm. nih.gov/geo/query/acc. cgi?acc=gse67462 | Publicly available at the NCBI Gene Expression Omnibus (accession no: GSE67462) |
| Wu J, Huang B, Chen H, Xie W | | 2016 | The landscape of accessible chromatin in mammalian pre-implantation embryos (ATAC-Seq) | https://www.ncbi.nlm. nih.gov/geo/query/acc. cgi?acc=GSE66581 | Publicly available at the NCBI Gene Expression Omnibus (accession no: GSE66581) |
| Lu F, Liu Y, Inoue A, Suzuki T, Zhao K, Zhang Y | | 2016 | Establishing Chromatin Regulatory Landscape during Mouse Preimplantation Development | https://www.ncbi.nlm. nih.gov/geo/query/acc. cgi?acc=GSE76642 | Publicly available at the NCBI Gene Expression Omnibus (accession no: GSE76642) |

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
