## [Decision Letter]

[Editors’ note: this article was originally rejected after discussions between the reviewers, but the authors were invited to resubmit after an appeal against the decision.]

Thank you for submitting your work entitled "The pioneer factor OCT4 requires BRG1 to functionally mature gene regulatory elements in mouse embryonic stem cells" for consideration by *eLife*. Your article has been favorably evaluated by a Senior Editor and three reviewers, one of whom, Irwin Davidson (Reviewer #1), is a member of our Board of Reviewing Editors.

Our decision has been reached after consultation between the reviewers. Based on these discussions and the individual reviews below, we regret to inform you that this version of your paper cannot be considered further for publication in *eLife*.

As can be seen from the reviews the referees considered that this was an interesting paper that addresses important questions concerning the regulation of pluripotency and the mechanisms of action of pioneer factors. The referees felt that the experimental system used was elegant and relevant for asking these important questions, yet quite a number of points still need further attention. In particular, it would be important to address the comments of referees 1 and 3 concerning nucleosome positioning with respect to OCT4 binding in presence or absence of BRG1 and provide some experiments to address the kinetics of chromatin reorganisation upon loss of OCT4 and the possible cooperative action of OCT4 with NANOG and SOX2 in BRG1 recruitment. In addition, as stressed by referee 2 the data analyses often lack appropriate statistical and quantitative aspects. Indeed referee 2 points to a number of statements in the text that are not backed up by quantitative analyses. Also discrepancies with previously published data need to be addressed and discussed.

While the large number of outstanding issues raised by the referees preclude simply revision of this version of the paper, we would be happy to consider in the future a new version that addresses the important points raised by the referees.

*Reviewer #1:*

The study by King and Klose describes the cooperation between OCT4 and BRG1 in establishing accessible chromatin domains that promote binding of OCT4 SOX2 and Nanog to regulatory elements of pluripotency expressed genes in mouse ES cells. They show that OCT4 recruits BRG1 to a large subset of its binding sites to establish an accessible chromatin state and that BRG1 is required for optimal OCT4 occupancy of many of its binding sites together with SOX2 and Nanog. Altogether the results are well presented and constitute an important insight into how OCT4 and BRG1 cooperate to promote transcription factor occupancy and pluripotency. Nevertheless several issues can be addressed.

The major outstanding issue in this study is mechanistically how do OCT4 and BRG1 render the chromatin accessible, what does accessibility really mean? The study uses ATAC-seq as a measure for chromatin accessibility, but as the paper describes OCT4-BRG1 cooperation it is maybe important to go one step further and analyse more precisely how this cooperation affects nucleosome positioning. Can the authors describe how loss of OCT4 or BRG1 affects nucleosome positioning? Can they show that OCT4 and BRG1 cooperate to generate a nucleosome-depleted region to which combinations of OCT4, SOX2 and Nanog bind? When either OCT4 or BRG1 are depleted does this affect nucleosome positioning in similar or different ways? According to the model of pioneer factors proposed in this paper, in absence of BRG1, OCT4 should bind regions occupied by nucleosomes that are displaced when BRG1 is present. Can the authors use the ATAC-seq data to map nucleosome positioning, if not perhaps they should perform Mnase mapping of nucleosome positioning in presence or absence of OCT4 and/or BRG1 to assess how positioning is altered and show that upon BRG1 recruitment critical nucleosomes are displaced to reveal the OCT4, SOX2 and Nanog binding sites.

In Figure 2 it seems that BRG1 binds at the center of the ATAC peak, but where does it bind relative to OCT4. Figure 4 gives the impression that OCT4 binding exactly co-localizes with BRG1 implying either that OCT4 binds on the nucleosome, or if OCT4 binds nucleosome depleted regions, then it means BRG1 does not bind the nucleosome, but binds OCT4? Can the authors comment on this and provide further insights into how the data should be interpreted?

*Reviewer #2:*

The authors use an elegant system to show that OCT4 leads to increases in chromatin accessibility in a manner partly dependent on BRG1. This leads to co-binding of another key ESC TF – SOX2. The authors further demonstrate that OCT4-BRG1 crosstalk is important for proper ESC gene expression. The manuscript is well written, well organized and clearly focuses on an important event in stem cell biology. An important conclusion from the data is that some pioneer factors need chromatin remodelers to exert their 'pioneering' activity and maintain pluripotency. However, it seems that the main novelty here is the genome-wide experiments when combined with the control of OCT4 and BRG1 expression in ESCs. Previous studies (cited in the manuscript) already show some aspects that are also presented in this study (link between OCT4 and BRG1, OCT4 leading to more accessible chromatin environment in multipotent cells, reliance of OCT4 on chromatin remodelers etc.).

1) The authors limit their study to the first 24 Hrs. of Brg1 knockdown, which restricts the ability to observe its actual effects on the maintenance of pluripotency. The authors make the conclusion that Brg1-bound enhancers regulate the pluripotency network. The authors should explore how Brg1 knock down affects pluripotency over the several days necessary to observe or not maintenance of ESCs or evidence of differentiation.

2) The authors use heatmaps throughout the manuscript. In the cases listed below, heatmaps are insufficient to support the claims raised by the authors. A quantitative analysis including statistics is needed.

A) Figure 1: This is an uncommon way of showing an association between gene expression and TF binding. The authors should try to link these two events in a manner which can be assessed statistically/quantitatively and not only visually (e.g. box plot, binned bar plot of number of sites as a function of distance from TSS etc.).

B) Figure 4: in the text (subsection 2 BRG1 supports transcription factor binding at distal gene regulatory elements”) the authors state that they found 'significant reductions in OCT4 binding at the majority (60%) of distal OCT4 targets following BRG1 removal'. How is that quantified? What is the cutoff criteria for calling a site BRG1-dependent or BRG1-independent? Heatmap analysis is insufficient to determine this.

C) Figure 4, subsection “Gene expression defects in the absence of BRG1 are linked to altered transcription factor binding”, see comment 2A.

D) Figure 5, subsection “BRG1-dependency reveals distinct modes of OCT4 function at distal regulatory elements during reprogramming and development”, last paragraph, see comment 2A.

3) Subsection “BRG1 is required to create accessible chromatin at OCT4 target sites in ESCs”. Figure 2—figure supplement 1: in the text the authors state that 'reductions in chromatin accessibility resulting from the loss of either OCT4 or BRG1 were 'highly similar'. However, the Venn diagram shows that only half of sites overlap. Moreover, the scatter plot seems to show only weak correlation. Importantly, the R^2^ is not presented so there is no way to evaluate the authors' statement.

4) Subsection “The pioneer factor OCT4 binds distal regulatory sites in pluripotent cells that would otherwise be inaccessible”, first paragraph. The authors need to provide more detail in the text and figure legends regarding the called Oct4 peaks (Figure 1) as well as what the exact chromatin signature is being used as a marker for distal regulatory elements (Figure 1). Some of the above is summarized in the Methods but it should also be detailed in the main text/legends. Also, does motif analyses show an Oct4 motif enriched at these binding sites? What percentage of sites contain the motif, how strong is the enrichment of the motif?

5) Subsection “The pioneer factor OCT4 binds distal regulatory sites in pluripotent cells that would otherwise be inaccessible”. The reference to "modest chromatin alterations" is ambiguous; a more specific description or quantitative comparison between these data and the cited papers is needed.

6) Subsection “BRG1 is required to create accessible chromatin at OCT4 target sites in ESCs”. In Figure 2, the authors need to show the Brg1 ChIP-seq heat map with Tamoxifen treatment as they did with Oct4 (+Dox) in Figure 1. Also, are these sites the same and in the same order as the Oct4 sites in Figure 1?

7) Subsection “OCT4 establishes chromatin accessibility by recruiting BRG1 to chromatin”. Figure 3: the correlation between the reduction in ATAC and BRG1 signals is weak (R^2^=0.33) and not 'very good' or 'high' as stated in the main text and the legend, respectively. This should be more carefully phrased. Also, the authors should reconcile the reason for the weak correlation.

8)Subsection 2 BRG1 supports transcription factor binding at distal gene regulatory elements”. In Figure 3, are the Oct4 Chip sites in the conditional Oct4 cell line (untreated) the same in the untreated conditional Brg1 cell line? Again, the authors should use the same sites as those shown in Figure 1. A scatter plot of the Oct4 ChIPs from these two cell lines would be good in a supplementary figure.

9) Subsections “BRG1 supports transcription factor binding at distal gene regulatory elements” and “Gene expression defects in the absence of BRG1 are linked to altered transcription factor binding”. What is meant by "functionally mature" TF binding events?

10) Subsection “BRG1-dependency reveals distinct modes of OCT4 function at distal regulatory elements during reprogramming and development”, last paragraph. In Figure 5, the differences between the Brg1 dependent and independent Oct4 sites are minimal over the development time frame.

*Reviewer #3:*

In this manuscript King and Klose describe that in mouse ES cells the binding of BRG1, a known as a subunit of the chromatin remodelling BAF complexes, is dependent on the presence of pluripotency factor OCT4. They also show that the presence of OCT4 at its binding sites is required for the binding of SOX2 and NANOG. In reverse experiments the authors find that the binding of OCT4 to the chromatin in ES cells is dependent on the presence of BRG1. From these and other experiments the authors conclude that their experiments reveal "a distinct requirement for a chromatin remodeller in shaping the activity of the pioneer factor OCT4 and regulating the pluripotency network".

1) From the presented experiments it seems that the BRG1/BAF complex is as much as a "pioneer" factor as is OCT4. In fact, the Schöler lab has shown (see Singhal et al. 2010) that the BAF complex is binding before OCT4 and Sox2 to the ES specific genes. Thus, the authors need to revise their ideas of OCT4 being a pioneer factor throughout the whole manuscript.

2) The authors should carry out ChIP-seq with a second subunit of the BAF complex to be able to show that the effects they see with BRG1 are indeed due to the BAF ATP dependent remodelling complex(es).

3) In the OCT4 knock down experiments the authors carry out ATAC-seq, Oct4, Sox2, Nanog and Brg1 ChIP-seqs, however in the BRG1 knock down conditions they carry out ATAC-seq, Oct4, Sox2, and Brg1 ChIP-seqs, but not Nonog ChIP-seq. Why? Is it possible that actually Nanog and may be Sox2 are recruiting Oct4 and Brg1/BAF complex to these sites?

4) To resolve these issues, the authors should carry out time scale experiments to test which factor(s) is leaving the first from the Oct4 occupied sites after Oct4 knock-down.

5) The analyses of Oct4 or Brg1 regulated genes in Figure 4 do not give the impression that these factors regulate the same subset of genes in spite of the fact that very often they seem to bind to the same sites. How can this be explained? Is it possible that Oct4 (or BRG1) are not regulating the analysed nearby genes?

---

## [Author Response]

[Editors’ note: the author responses to the first round of peer review follow.]

*[…] Reviewer #1:*

*[…] The major outstanding issue in this study is mechanistically how do OCT4 and BRG1 render the chromatin accessible, what does accessibility really mean? The study uses ATAC-seq as a measure for chromatin accessibility, but as the paper describes OCT4-BRG1 cooperation it is maybe important to go one step further and analyse more precisely how this cooperation affects nucleosome positioning. Can the authors describe how loss of OCT4 or BRG1 affects nucleosome positioning? Can they show that OCT4 and BRG1 cooperate to generate a nucleosome-depleted region to which combinations of OCT4, SOX2 and Nanog bind? When either OCT4 or BRG1 are depleted does this affect nucleosome positioning in similar or different ways? According to the model of pioneer factors proposed in this paper, in absence of BRG1, OCT4 should bind regions occupied by nucleosomes that are displaced when BRG1 is present. Can the authors use the ATAC-seq data to map nucleosome positioning, if not perhaps they should perform Mnase mapping of nucleosome positioning in presence or absence of OCT4 and/or BRG1 to assess how positioning is altered and show that upon BRG1 recruitment critical nucleosomes are displaced to reveal the OCT4, SOX2 and Nanog binding sites.*

We agree with the reviewer it is useful to go a step further and define more precisely how the cooperation between OCT4 and BRG1 affects nucleosome occupancy and positioning at OCT4 target sites. Therefore, we have now performed a new set of analyses on the OCT4^cond^ and Brg1^fl/fl^ ATAC-seq datasets using the NucleoATAC analysis approach (Schep et al., 2015) to measure nucleosome signal in the presence or absence of OCT4/BRG1. In wild type cells, a nucleosome-depleted region is clearly evident at OCT4 binding sites, as demonstrated by the reduced nucleosome signal at the centre of the OCT4 peak compared to neighbouring regions (Figure 1 and Figure 4). Importantly, OCT4 and the BRG1/BAF complex are enriched within this nucleosome-depleted site (also see response to Point 2, and Figure 4). We then compared how removal of OCT4 or BRG1 affected the nucleosome-depleted state at OCT4 target sites. This analysis revealed widespread increases in nucleosome signal at sites where OCT4 binding is required for chromatin accessibility following OCT4 removal (Figure 1). In contrast, OCT4 target sites where OCT4 is not required for accessibility (i.e. OCT4-independent) (Figure 1) remained depleted of nucleosome signal, suggesting that chromatin accessibility measured by ATAC-seq may directly reflect the underlying nucleosome occupancy. Therefore, OCT4-dependent processes play a central role at the majority of OCT4 binding sites in shaping accessibility and displacing nucleosomes. Importantly, when we examined changes in nucleosome signal in the BRG1-depleted cells at OCT4 bound sites (Figure 3, Figure 3—figure supplement 1) there was no longer the same extent of nucleosome depletion. This is consistent with BRG1 cooperating with OCT4 to establish chromatin accessibility/nucleosome depletion at OCT4 target sites. Intriguingly, sites that retained OCT4 binding in the absence of BRG1 (BRG1-independent OCT4 targets) remained depleted of nucleosomes (Figure 5—figure supplement 2), suggesting that BRG1-independent activities function to define accessibility and nucleosome-depletion at these sites. Together, these observations demonstrate that BRG1-dependent OCT4 targets (which represent the majority of OCT4 sites) rely on both OCT4 binding and BRG1 to create accessible chromatin and to displace nucleosomes.

We now describe these important new observations in the subsections “The pioneer factor OCT4 binds distal regulatory sites in pluripotent cells that would otherwise be inaccessible”, last paragraph, “BRG1 is required to create accessible chromatin at OCT4 target sites in ESCs”, and “BRG1 supports transcription factor binding at distal gene regulatory elements”.

*In Figure 2 it seems that BRG1 binds at the center of the ATAC peak, but where does it bind relative to OCT4. Figure 4 gives the impression that OCT4 binding exactly co-localizes with BRG1 implying either that OCT4 binds on the nucleosome, or if OCT4 binds nucleosome depleted regions, then it means BRG1 does not bind the nucleosome, but binds OCT4? Can the authors comment on this and provide further insights into how the data should be interpreted?*

We have now examined the localisation of OCT4 and BRG1 in more detail using high-resolution metaplots (10bp resolution) and compared this to nucleosome signal at OCT4-bound sites (see point 1). This analysis revealed that OCT4 and BRG1/BAF co-localise within a very narrow nucleosome-depleted region at the centre of OCT4 binding sites (Figure 4). As suggested by the reviewer, this appears to be consistent with OCT4-dependent recruitment of BRG1 to its target sites. The fact that OCT4 can bind nucleosomal DNA (Soufi et al., 2012, Soufi et al., 2015) and is responsible for recruiting BRG1 to create accessible and nucleosome-depleted chromatin at most of its target sites (Figure 3 and Figure 4) is also in fitting with the assisted loading model for transcription factor binding (Voss and Hager, 2014, Swinstead et al., 2016). This model invokes a process whereby transcription factor-dependent binding to nucleosomal DNA (i.e. OCT4) supports recruitment of remodelling activities (i.e. BRG1) to create accessible chromatin that supports further transcription factor binding/co-binding (OCT4/SOX2/NANOG) events. However, we would like to point out that despite the apparent nucleosome-depletion at OCT4 bound sites, we cannot exclude the possibility that OCT4/BRG1 remains bound to an unstable nucleosome during the process of creating local accessibility at target sites (Ishii et al., 2015, Voong et al., 2016). Nevertheless, the high degree of co-localization of OCT4 and BRG1 is a consistent model of OCT4-dependent recruitment of BRG1 to chromatin, where it appears necessary to maintain a nucleosome-depleted state at OCT4 target sites (see response to Point 1). We have emphasized and clarified these points in the subsection “OCT4 establishes chromatin accessibility by recruiting BRG1 to chromatin”.

*Reviewer #2:*

*The authors use an elegant system to show that OCT4 leads to increases in chromatin accessibility in a manner partly dependent on BRG1. This leads to co-binding of another key ESC TF – SOX2. The authors further demonstrate that OCT4-BRG1 crosstalk is important for proper ESC gene expression. The manuscript is well written, well organized and clearly focuses on an important event in stem cell biology. An important conclusion from the data is that some pioneer factors need chromatin remodelers to exert their 'pioneering' activity and maintain pluripotency. However, it seems that the main novelty here is the genome-wide experiments when combined with the control of OCT4 and BRG1 expression in ESCs. Previous studies (cited in the manuscript) already show some aspects that are also presented in this study (link between OCT4 and BRG1, OCT4 leading to more accessible chromatin environment in multipotent cells, reliance of OCT4 on chromatin remodelers etc.).*

We thank the reviewer for their kind words regarding our manuscript and we agree that our study addresses an important event in stem cell biology. As pointed out by the reviewer, several previous studies have touched on aspects of OCT4 and BRG1 function related to our study. However, in many cases these studies have focused on a small number of genes, have been performed in different pluripotent cell types (ESCs, embryonal carcinoma, iPSCs), and often come to differing conclusions making it difficult to grasp the scale, validity, and relevance of these often isolated observations. For example, it has remained elusive how OCT4 engages with the thousands of distal regulatory sites it binds in the ESC genome and how this is related to its proposed role as a pioneer transcription factor. In particular, previous studies examining the relationship between OCT4 binding and chromatin accessibility/nucleosome positioning have not addressed how OCT4 transitions inaccessible chromatin into functionally active and accessible regulatory elements required for gene regulation (see response to Point 5 for details on these studies). Furthermore, the link between BRG1 and OCT4 has largely stemmed from simple biochemical immunoprecipitation analysis (Pardo et al., 2010, van den Berg et al., 2010, Ding et al., 2012), with the functional relevance of this proposed interaction remaining unclear.

Here we have taken a systematic and genome-wide approach to address how OCT4 engages with the genome in mouse ESCs and demonstrate a defined role for BRG1 in shaping transcription factor binding, chromatin accessibility and gene regulatory function. We believe this encompasses an important step forward for understanding OCT4 function in ESCs. Furthermore, we provide new evidence that OCT4s function as a pioneer factor relies on an assisted loading type mechanism that utilizes the function of chromatin remodelling machines, which to our knowledge has not been previously proposed. We believe these important and novel discoveries will be well received and highly cited by the readership of *eLife*.

*1) The authors limit their study to the first 24 Hrs. of Brg1 knockdown, which restricts the ability to observe its actual effects on the maintenance of pluripotency. The authors make the conclusion that Brg1-bound enhancers regulate the pluripotency network. The authors should explore how Brg1 knock down affects pluripotency over the several days necessary to observe or not maintenance of ESCs or evidence of differentiation.*

A central goal of our study was to understand in molecular detail, at the genome-scale, how OCT4 interacts with and functions on the genome to orchestrate the pluripotent ESC state, something that in our view is still poorly understood (which we now highlight in the subsection “BRG1 is required to create accessible chromatin at OCT4 target sites in ESCs”). In doing so, we have purposely limited our analysis to time points following genetic ablation of OCT4 and BRG1 that are sufficient to allow protein loss, but where cells still exist in a state that is morphologically and biochemically ESC-like (Figure 1 and Figure 3). Our rationale for exploiting this approach was as a means to define the molecular mechanisms that support how OCT4 and other pluripotency-associated transcription factors (SOX2 and NANOG) recognize and regulate the pluripotent ESC genome. This revealed that BRG1 is required for normal OCT4/SOX2/NANOG engagement with the ESC genome and to functionally mature distal regulatory elements in controlling gene expression. As the reviewer correctly points out, these observations are important for understanding how the pluripotent state is lost after prolonged BRG1 depletion, which has been the focus of previous studies (Ho et al., 2009, Kidder et al., 2009, Zhang et al., 2014). Therefore, in the context of our defined questions about the molecular mechanisms by which OCT4 and BRG1 cooperate to interact with and regulate the pluripotent genome, it seems unlikely that additional insight will be gained from extending our analysis to later time points where differentiation has ensued.

2) The authors use heatmaps throughout the manuscript. In the cases listed below, heatmaps are insufficient to support the claims raised by the authors. A quantitative analysis including statistics is needed.

We agree that heatmaps by themselves do not provide any quantitative or statistical support. However, they do provide an intuitive way for non-specialists who will read the manuscript in *eLife* to appreciate the striking reductions and trends in our ATAC-seq, ChIP-seq and RNA-seq datasets. Importantly, in parallel to these heatmap visualisations, we performed robust statistical analyses on all of the datasets presented in heatmaps in the original version of the manuscript. For example, all genomic experiments were performed in biological triplicate and differential transcription factor binding or accessibility was identified and statistically validated using the DiffBind package (Stark and Brown, 2011), with significance being achieved when a FDR < 0.05 and a fold change > 1.5-fold was realized (detailed in the subsection “Differential binding and gene expression analysis” in the Materials and methods). We acknowledge that these statistical measures were not highlighted sufficiently in the original submission, and have therefore included additional analyses and figures throughout the revised manuscript that provide a more detailed and quantitative interrogation of the genomic data. While these analyses have undoubtedly increased the statistical robustness of our study, we would like to stress that our interpretation of the data remains essentially unchanged from the original manuscript. Precise details of these additional analyses and figures are included below.

*A) Figure 1: This is an uncommon way of showing an association between gene expression and TF binding. The authors should try to link these two events in a manner which can be assessed statistically/quantitatively and not only visually (e.g. box plot, binned bar plot of number of sites as a function of distance from TSS etc.).*

We have now carried out additional quantitative and statistical analysis to reinforce the observation that reductions in chromatin accessibility and transcription factor binding associated with the loss of OCT4 is linked to reduced expression of nearby genes (see Figure 2 in revised manuscript). Firstly, we separated the OCT4-bound sites into those that require OCT4 for accessibility (OCT4-dependent) and those that do not (OCT4-independent) (DiffBind FDR < 0.05; fold change > 1.5). We then used box plots to examine the log2 fold changes in expression of the closest genes (Figure 2). This revealed that genes in close proximity to OCT4-dependent regulatory elements are significantly more likely to exhibit reductions in gene expression than genes located near OCT4 bound sites that do not rely on OCT4 for accessibility (p < 7.6 × 10^-43^). Secondly, we identified genes that are significantly down-regulated in the OCT4-depleted ESCs (DESeq2 FDR < 0.05; fold change > 1.5). We then compared the OCT4 target sites for which the nearest gene shows such a significant reduction in expression (OCT4-dependent expression) with the OCT4 targets sites that show significant reductions in accessibility (OCT4-dependent accessibility). Through this analysis, we observe that the majority of significant reductions in gene expression are associated with significant reductions in chromatin accessibility at nearby OCT4 target sites (Figure 2). As expected, many OCT4-dependent accessibility changes were not associated with changes in gene expression, which may reflect redundancy with other distal regulatory elements, regulation of non-proximal genes, or non-functional OCT4 binding sites. Finally, to examine whether the loss of chromatin accessibility at distal OCT4 bound sites is linked to expression changes of nearby genes, we compared the distance to the nearest down-regulated TSS from OCT4 peaks with or without OCT4-dependent accessibility (Figure 2). This nicely illustrated that OCT4-bound regulatory elements that lose accessibility following removal of OCT4 are closer (median 16.2 kb) to downregulated genes than those that do not rely on OCT4 for accessibility (median 125.4 kb) (p < 4.2 × 10^-147^). Together these new statistical and quantitative analysis support our previous conclusions that OCT4 plays an essential role in determining regulatory element accessibility and function, which is important for the normal expression of nearby genes associated with the pluripotency transcriptional network.

The description and discussion of these new analyses is included in the last paragraph of the subsection “OCT4 supports transcription factor binding at distal regulatory elements to regulate pluripotency-associated genes”.

*B) Figure 4: in the text (subsection 2 BRG1 supports transcription factor binding at distal gene regulatory elements”) the authors state that they found 'significant reductions in OCT4 binding at the majority (60%) of distal OCT4 targets following BRG1 removal'. How is that quantified? What is the cutoff criteria for calling a site BRG1-dependent or BRG1-independent? Heatmap analysis is insufficient to determine this.*

To identify and quantify sites that have significant reductions in OCT4 binding following BRG1 removal we used the DiffBind package with a significant change in OCT4 binding being achieved when a FDR < 0.05 and a fold change > 1.5-fold was observed from biological triplicate experiments. This statistical analysis revealed that 60% of OCT4 bound distal sites showed significant reductions in OCT4 binding following removal of BRG1. In addition to stating this value in text, we have now also included a pie chart for this and similar statistical analysis of SOX2 and NANOG binding in Figure 5that better convey this observation.

However, we were conscious in reflection that one issue associated with using this binary segregation of affected/unaffected sites is that it relies on a somewhat arbitrary user defined thresholds (FDR < 0.05 and a fold change > 1.5-fold) which can lead to a range of affected sites that fall just below the fold change threshold being considered unaffected (false negatives). In subsequent visualisation and analysis this can cloud one’s appreciation of the features associated with BRG1 dependency. To overcome this limitation and focus on the clearest BRG1 dependencies, in the revised manuscript we have stratified OCT4 targets into the 20% of sites that were least dependent on BRG1 (BRG1-independent) and compared these to the 20% of sites that were most dependent on BRG1 (BRG1-dependent). This now allows the reader to more clearly appreciate how BRG1 contributes to a series of features associated with OCT4 target sites in Figure 5 and Figure 6.

To clarify this for the reader we have highlighted the distinction between these two strategies in the figure legends and in the subsection “BRG1 supports transcription factor binding at distal gene regulatory elements”.

*C) Figure 4, subsection “Gene expression defects in the absence of BRG1 are linked to altered transcription factor binding”, see comment 2A.*

We have now further explored the relationship between altered transcription factor binding and gene expression in the BRG1-depleted ESCs. In new analyses included in Figure 5 and Figure 5—figure supplement 4 we have examined the expression changes of genes in close proximity to OCT4 bound sites that have different requirements for BRG1 in OCT4 occupancy (Figure 5) and quantified the trend observed in the original heatmap (previously Figure 4, now Figure 5). This revealed that sites that lose OCT4 are more likely to experience reductions in expression of nearby genes (Figure 5). In contrast, sites that retain or display increases in OCT4/SOX2/NANOG (BRG1-independent sites; see metaplot in Figure 5) are associated with increases in gene expression (Figure 5). Furthermore, we have quantified the changes in transcription factor binding at distal OCT4 target sites (box plots) in close proximity to the genes with altered gene expression (Figure 5), and confirm that genes with reduced expression are associated with the largest reductions in transcription factor binding at nearby OCT4 target sites. We have also included several genomic snapshots depicting RNA-seq and transcription factor ChIP-seq for exemplary genes which experience reduced, unchanged or increased expression (Figure 5—figure supplement 4). Together these analyses provide additional support to our previous conclusions drawn from the heatmap visualisation that the inability of OCT4 and other transcription factors to interact normally with their target sites in the absence of BRG1 is reflected in changes in nearby gene expression. These additional analyses are presented and discussed in the last paragraph of the subsection “Gene expression defects in the absence of BRG1 are linked to altered transcription factor binding”.

*D) Figure 5, subsection “BRG1-dependency reveals distinct modes of OCT4 function at distal regulatory elements during reprogramming and development”, last paragraph, see comment 2A.*

In Figure 5 (now Figure 6 in the revised manuscript), we describe that genes associated with BRG1-dependent OCT4 binding sites appear to be activated later during iPSC reprogramming than genes associated with BRG1-independent OCT4 targets. To extend and strengthen this initial observation, we have performed new quantitative analysis to replace the heatmap visualisation, which we agree was difficult to interpret. In this new analysis (Figure 6), we have examined the activation dynamics during iPSC generation of genes that require OCT4 for their normal expression in embryonic stem cells (OCT4-dependent genes), and compared genes associated with BRG1-dependent or BRG1-independent OCT4 binding. By quantifying log2FC of gene expression, we demonstrate that genes associated with BRG1-independent OCT4 binding are activated earlier during reprogramming than genes associated with BRG1-dependent OCT4 binding, in fitting with our original observations. Statistical confidence is provided by the 95% confidence intervals visualised on these plots, and we have now performed this analysis on two independent iPSC reprogramming RNA-seq datasets (Chen et al., 2016, Cieply et al., 2016). This provides new quantitative support that BRG1-independent OCT4 binding is associated with early gene activation during reprogramming, while genes associated with BRG1-dependent OCT4 binding are activated later.

*3) Subsection “BRG1 is required to create accessible chromatin at OCT4 target sites in ESCs”. Figure 2—figure supplement 1: in the text the authors state that 'reductions in chromatin accessibility resulting from the loss of either OCT4 or BRG1 were 'highly similar'. However, the Venn diagram shows that only half of sites overlap. Moreover, the scatter plot seems to show only weak correlation. Importantly, the R^[3]^ is not presented so there is no way to evaluate the authors' statement.*

Having identified a link between BRG1 binding and OCT4-dependent chromatin accessibility (Figure 3), it was imperative that we directly examine whether BRG1 contributed to chromatin accessibility at OCT4 target sites. Genomic snapshots (Figure 3, Figure 3—figure supplement 1), heatmap visualisation (Figure 3), and new metaplot visualisation (Figure 3—figure supplement 1) illustrate that BRG1 clearly plays a widespread role in maintaining normal chromatin accessibility at OCT4 bound sites. When we compared the significant changes (DiffBind FDR < 0.05; fold change > 1.5) in chromatin accessibility following BRG1 removal with the significant changes in chromatin accessibility following OCT4 removal, there was a substantial, but not complete overlap between OCT4-dependent accessibility and BRG1-dependent accessibility as highlighted by the reviewer. However, this Venn diagram analysis was based on sites that had statistically significant alterations in binding that were constrained by user defined fold change thresholds. One limitation of this type of approach is that fold change thresholds can overlook affected sites that fall below the user defined threshold (false negatives) as discussed above in reviewer 2 – Point 2B. Therefore, the original Venn diagram may not have done justice to the breadth of sites that show related alterations in accessibility following OCT4 and BRG1 removal. Therefore, to examine this relationship in more detail we have now carried out unbiased clustering analysis of OCT4-dependent sites based on log2FC ATAC signal in the Brg1^fl/fl^ and OCT4^cond^ cells. This illustrates that chromatin accessibility at 76% of OCT4 targets sites relies on BRG1 (see cluster 1, Figure 3—figure supplement 1). Furthermore, even sites which are classified as BRG1-independent, still display some reductions in ATAC-seq signal (see cluster 2, Figure 3—figure supplement 1), consistent with decreased chromatin accessibility at OCT4 targets sites in the absence of BRG1.

However, it is clear that the reduction in ATAC-seq signal observed at individual OCT4 target sites following BRG1 removal does not always correlate precisely with the reduction in ATAC-seq signal following OCT4 removal. This is evident from the scatterplot in Figure 3—figure supplement 1 (with R^2^ and cor values now included). However, this is actually what we expected based on the heatmap visualisation, as it is clear that not all ATAC-seq signal at OCT4 target sites results directly from BRG1 activity. We suspect that this is due in part to transcription factor binding which is retained at some sites following BRG1 depletion (Figure 5). Nevertheless, our previous and new analysis clearly demonstrates a widespread role for OCT4 in targeting BRG1 in ESCs (Figure 4), with the majority of these sites requiring BRG1 to maintain normal chromatin accessibility (Figure 3).

We have included these new analyses into a new supplementary figure (Figure 3—figure supplement 1) and discuss the above points in the subsection “BRG1 is required to create accessible chromatin at OCT4 target sites in ESCs”

*4) Subsection “The pioneer factor OCT4 binds distal regulatory sites in pluripotent cells that would otherwise be inaccessible”, first paragraph. The authors need to provide more detail in the text and figure legends regarding the called Oct4 peaks (Figure 1) as well as what the exact chromatin signature is being used as a marker for distal regulatory elements (Figure 1). Some of the above is summarized in the Methods but it should also be detailed in the main text/legends.*

As suggested by the reviewer we have now clarified the chromatin signature used to annotate OCT4 binding sites as either distal regulatory element or promoters. This description is now included in both the main text and in the figure legend (see subsections “The pioneer factor OCT4 binds distal regulatory sites in pluripotent cells that would otherwise be inaccessible”, first paragraph, and “Functional annotation of transcription factor binding sites”). Briefly, this was achieved by examining the relative enrichment of H3K4me3 (usually associated with promoters) and H3K4me1 (usually associated with distal regulatory elements) at OCT4 bound sites as previously described (Hay et al., 2016). To illustrate the effectiveness of these segregations, we have carried out additional metaplot visualisation of H3K4 modifications and calculated the distance to nearest annotated TSS for each class of site (Figure 1—figure supplement 1).

Also, does motif analyses show an Oct4 motif enriched at these binding sites? What percentage of sites contain the motif, how strong is the enrichment of the motif?

Our high confidence OCT4 interval set was generated by identifying OCT4 peaks (using DANPOS2) that lose OCT4 ChIP-seq signal after removal of OCT4. To our knowledge, no other previous OCT4 ChIP-seq analysis has included this important control that ensures that the OCT4 ChIP-seq signal is due to the protein and not due to other non-specific ChIP-seq signal. As suggested by the reviewer we have further characterised this interval set by performing motif enrichment analysis (for both canonical motifs and de novo motif enrichment) at our 15,920 OCT4 peaks using the MEME suite of tools. As expected, this analysis revealed a striking enrichment of the OCT4:SOX2 dual motif sequence, in addition to significant enrichments for the highly similar OCT2 motif and two de novo OCT4 motifs. These motifs are enriched at the peak centre (CentriMO analysis), and 60% of peaks contained at least one of these motifs. This is comparable with other studies for transcription factor motifs and binding (Valouev et al., 2008, Heinz et al., 2010). We have included these new analyses in a new supplementary figure (Figure 1—figure supplement 1) and describe this in the subsections “The pioneer factor OCT4 binds distal regulatory sites in pluripotent cells that would otherwise be inaccessible”, first paragraph and “Differential binding and gene expression analysis”.

*5) Subsection “The pioneer factor OCT4 binds distal regulatory sites in pluripotent cells that would otherwise be inaccessible”. The reference to "modest chromatin alterations" is ambiguous; a more specific description or quantitative comparison between these data and the cited papers is needed.*

In this line of the text we are referring to previous studies that have proposed a role for OCT4 in promoting alterations to chromatin but which were limited in their scope, technical approach, rigor or interpretation. For example, Shakya et al. (2015) performed histone H3 ChIP-qPCR at a single OCT4 target (the POU5F1 enhancer) and observed depletion of H3 signal after OCT4 over-expression in differentiated cells. You et al. (2011) observed increased nucleosome occupancy (as measured by M.CviPI digestion efficiency) at two OCT4 targets after OCT4 siRNA in NCCIT embryonic carcinoma cells. In both of these cases, the effects on chromatin accessibility/nucleosome occupancy are appreciable and significant, but these studies are limited in that they examine individual loci and are not directly comparable to our study in ESCs.

Alternatively, two studies have generated genome-wide datasets examining the role of OCT4 in determining chromatin accessibility using different approaches. Chen et al. (2014) perform restriction endonuclease digestion (RED)-seq 5 days after knocking down OCT4 with siRNA in mouse ESCs and observed reduced chromatin accessibility at regulatory elements genome-wide. However, ESCs cells depleted of OCT4 for extended periods of time (i.e. 5 days) differentiate (Niwa et al., 2000), meaning that these observations are of little relevance to understanding the precise contribution of OCT4 to maintenance of the pluripotent state that is the focus of our study. Furthermore, these experiments were only performed in biological singlicate, meaning that interpretations drawn from any comparison with this data would lack experimental, statistical, and quantitative rigor. Lu et al. (2016) perform DNase-seq in the early mouse embryo after ablating OCT4 expression with siRNA at the 8-cell stage. However, as described in the original manuscript, knockdown of OCT4 results in reduced DNase-seq signal at both OCT4-bound and OCT4-negative regulatory elements. We therefore find it difficult to interpret whether the observed effects are driven by OCT4 or are secondary effects associated with alterations in cellular trajectory that would result from removal of OCT4 at this early developmental stage. For these reasons, we find it difficult to make meaningful and quantitative comparisons of these studies with our data. In fact, these studies exemplify a lack of clarity in the current understanding of OCT4s function in shaping chromatin structure and accessibility in the pluripotent state, which was one of the central motivations for our detailed and systematic genome-wide study in ESCs.

Nevertheless, in order to better reflect the conclusions drawn by these previous studies, we have rephrased the sentence originally highlighted by the reviewer with the following text, although do not describe each in detail each for the sake of brevity.

“This is supported by previous studies describing a role for OCT4 in maintaining nucleosome-depleted regions and/or chromatin accessibility at individual loci in pluripotent cells (You et al., 2011, Shakya et al., 2015) or genome-wide (Chen et al., 2014, Lu et al., 2016).”

*6) Subsection “BRG1 is required to create accessible chromatin at OCT4 target sites in ESCs”. In Figure 2, the authors need to show the Brg1 ChIP-seq heat map with Tamoxifen treatment as they did with Oct4 (+Dox) in Figure 1. Also, are these sites the same and in the same order as the Oct4 sites in Figure 1?*

We have now performed BRG1 ChIP-seq in the Brg1^fl/fl^ ESCs before and after tamoxifen treatment and have visualised these data by heat mapping as suggested by the reviewer. We have also included genomic snapshots for this BRG1 ChIP-seq data in Figure 3 and Figure 3—figure supplement 1. As expected, treatment of the Brg1^fl/fl^ ESCs with tamoxifen results in dramatic reduction in BRG1 ChIP-seq signal at OCT4 target sites. Furthermore, these OCT4 target sites are ranked in the same order as in Figure 1. We have adjusted the Figure legend to indicate this.

*7) Subsection “OCT4 establishes chromatin accessibility by recruiting BRG1 to chromatin”. Figure 3: the correlation between the reduction in ATAC and BRG1 signals is weak (R^2^=0.33) and not 'very good' or 'high' as stated in the main text and the legend, respectively. This should be more carefully phrased. Also, the authors should reconcile the reason for the weak correlation.*

In Figure 3 (now Figure 4), we have provided two different measures of statistical confidence, the linear regression co-efficient (R^2^), which represents the goodness of fit between the two variables and Pearson correlation coefficient (cor) to quantify the degree to which these two variables are related. Therefore, the Pearson correlation is 0.57 which we believe represents a good correlation between these two variables given that we are comparing two different genomic assays (ChIP-seq and ATAC-seq) that have very different inherent features (i.e. dynamic range and signal to noise). Furthermore, we have now performed ChIP-seq for an additional BAF subunit (SS18), and this has revealed similar trend but with an appreciably higher correlation (cor = 0.80). We suspect this may reflect the better signal-to-noise of the SS18 ChIP-seq compared to the BRG1 ChIP-seq. We have included this new analysis in the manuscript (Figure 4) and have adjusted the language we use in the main text and figure legend to reflect the nature of this correlation more accurately in the subsection “OCT4 establishes chromatin accessibility by recruiting BRG1 to chromatin”.

*8) Subsection “BRG1 supports transcription factor binding at distal gene regulatory elements”. In Figure 3, are the Oct4 Chip sites in the conditional Oct4 cell line (untreated) the same in the untreated conditional Brg1 cell line? Again, the authors should use the same sites as those shown in Figure 1. A scatter plot of the Oct4 ChIPs from these two cell lines would be good in a supplementary figure.*

Throughout the manuscript we have used a single OCT4 peak set based on the OCT4 ChIP-seq performed in the OCT4^cond^ ESCs, which is described and characterised in Figure 1 and Figure 1—figure supplement 1. We have used this peak set in Figure 1–Figure 4, and a subset of these intervals (only distal regulatory element peaks) for Figure 5 and Figure 6 where our analysis is focused on distal regulatory elements. As suggested by the reviewer we have compared the wild type OCT4 ChIP-seq from the two different cell lines using a scatterplot and regression/correlation analysis. This confirmed that OCT4 binding is highly similar in the two cell lines. This has now been included in Figure 5—figure supplement 1 and is described in the subsection “BRG1 supports transcription factor binding at distal gene regulatory elements”.

*9) Subsections “BRG1 supports transcription factor binding at distal gene regulatory elements” and “Gene expression defects in the absence of BRG1 are linked to altered transcription factor binding”. What is meant by "functionally mature" TF binding events?*

We have used this term to describe the process by which the initial sampling or binding of OCT4 to inaccessible chromatin is ultimately resolved into the formation of distal regulatory elements with a full complement of transcription factor binding and gene regulatory capacity. Importantly our work demonstrates a clear role for BRG1/BAF in the functional maturation of OCT4 target sites in ESCs. We have clarified this point at various points throughout the manuscript to better reflect this (see subsections “BRG1 supports transcription factor binding at distal gene regulatory elements” and “Gene expression defects in the absence of BRG1 are linked to altered transcription factor binding”, last paragraph).

*10) Subsection “BRG1-dependency reveals distinct modes of OCT4 function at distal regulatory elements during reprogramming and development”, last paragraph. In Figure 5, the differences between the Brg1 dependent and independent Oct4 sites are minimal over the development time frame.*

In Figure 5 (now Figure 6 in the revised manuscript), we illustrate that sites that rely on BRG1 for OCT4 binding in ESCs gain accessibility later than BRG1-independent OCT4 binding sites during early mouse development. We have now performed a quantitative analysis that illustrates this point more clearly in Figure 6 and Figure 6—figure supplement 1. Firstly, in Figure 6 we have quantified and plotted the ATAC-seq read density for BRG1-dependent or BRG1-independent OCT4-bound distal regulatory elements as development proceeds to more easily visualise the gains in chromatin accessibility depicted in the metaplots in Figure 6. Statistical confidence is provided by the 95% confidence intervals for each group. Secondly, we have performed the identical analysis on a comparable but independently generated DNase-seq dataset in the early mouse embryo (Figure 6—figure supplement 1; Lu et al., 2016). This revealed the same trend as Figure 6, strengthening our original observation that BRG1-independent targets become accessible earlier, and BRG1-dependent later, during early mouse development and reprogramming.

*Reviewer #3:*

[…] 1) From the presented experiments it seems that the BRG1/BAF complex is as much as a "pioneer" factor as is OCT4. In fact, the Schöler lab has shown (see Singhal et al. 2010) that the BAF complex is binding before OCT4 and Sox2 to the ES specific genes. Thus, the authors need to revise their ideas of OCT4 being a pioneer factor throughout the whole manuscript. #

From our reading of the Singhal et al. (2010) study we do not find evidence to support the BAF complex functioning as a pioneer factor at target sites prior to OCT4 and SOX2 binding. In fact, binding of the BAF complex was not examined in this study (by ChIP or otherwise). Instead, what the Singhal et al. study demonstrates is that overexpression of BRG1/BAF with the reprogramming factors OCT4, KLF, and SOX2 (OKS) in mouse embryonic fibroblasts increases the efficiency of reprogramming and iPSC formation (now described in the second paragraph of the Discussion). While this clearly indicates that BRG1 is important for the efficiency of OCT4-driven reprogramming, it does not suggest that BRG1 functions as a pioneer factor. This term is usually reserved to describe sequence-specific transcription factors that are capable of binding inaccessible chromatin to establish new functional regulatory elements and drive gene expression (see Magnani et al., 2011, Iwafuchi-Doi and Zaret, 2014, Swinstead et al., 2016). Based on our observations it seems more plausible that the iPSC reprogramming process is potentiated by BAF through its recruitment to regulatory sites via OCT4 in a temporally coincident manner. Indeed, similar conclusions were arrived at in a recently published study examining BRG1 binding during reprogramming (Chronis et al., 2017), which we now cite in the subsection “OCT4 establishes chromatin accessibility by recruiting BRG1 to chromatin”. In our view, these observations would exclude BRG1/BAF from being considered as a classical pioneer transcription factor, but instead are consistent with OCT4 recruiting BRG1 to distal regulatory elements as part of an assisted loading mechanism for pioneer transcription factor binding and function as suggested in our original submission.

*2) The authors should carry out ChIP-seq with a second subunit of the BAF complex to be able to show that the effects they see with BRG1 are indeed due to the BAF ATP dependent remodelling complex(es).*

As suggested by the reviewer we have now performed ChIP-seq for SS18, which forms part of the BAF remodelling complex in ESCs and other cell types (Kadoch and Crabtree, 2013), before and after removal of OCT4. In agreement with our BRG1 ChIP-seq analysis, SS18 binding is largely dependent upon OCT4 at OCT4 bound distal regulatory sites (Figure 4, subsection “OCT4 establishes chromatin accessibility by recruiting BRG1 to chromatin”). Furthermore, the reductions in both BAF subunits correlated with the alterations in chromatin accessibility observed by ATAC-seq (Figure 4). This provides new evidence supporting the conclusion that OCT4 recruits the BAF remodelling complexes to chromatin in order to facilitate chromatin accessibility and transcription factor binding.

*3) In the OCT4 knock down experiments the authors carry out ATAC-seq, Oct4, Sox2, Nanog and Brg1 ChIP-seqs, however in the BRG1 knock down conditions they carry out ATAC-seq, Oct4, Sox2, and Brg1 ChIP-seqs, but not Nonog ChIP-seq. Why?*

As suggested by the reviewer we have now performed NANOG ChIP-seq in the Brg1^fl/fl^ ESCs before and after tamoxifen treatment. The effect on NANOG binding following removal of BRG1 is highly similar to SOX2, with sites relying on BRG1 for OCT4 binding displaying reductions NANOG occupancy and sites that retain OCT4 binding displaying maintenance or even increased in NANOG occupancy (Figure 5). We have now described these observations in the subsection “BRG1 supports transcription factor binding at distal gene regulatory elements”.

*Is it possible that actually Nanog and may be Sox2 are recruiting Oct4 and Brg1/BAF complex to these sites?*

It is likely that SOX2 contributes to the occupancy of OCT4 at its binding sites given the previously described co-operative and synergistic DNA binding by OCT4 and SOX2 in vitro (Reményi et al., 2003). However, OCT4 can bind to its target sites on nucleosomal DNA independently of SOX2 (Soufi et al., 2015), suggesting that binding with SOX2 may function to stabilise OCT4 binding as opposed to directing its initial binding. Consistent with these in vitro observations, our in vivo experiments suggest that SOX2 or NANOG are not sufficient to recruit BRG1 as SOX2 and/or NANOG are retained at some target sites following depletion of OCT4, yet BRG1 binding is either completely lost or substantially reduced (Figure 7).

Author response image 1.BRG1/BAF occupancy relies upon OCT4 but not SOX2 and NANOG binding.Genomic snapshots of OCT4 targets that retain SOX2 and NANOG binding in the absence of OCT4, but lose BRG1 and SS18 ChIP-seq signal.**DOI:**
http://dx.doi.org/10.7554/eLife.22631.019

*4) To resolve these issues, the authors should carry out time scale experiments to test which factor(s) is leaving the first from the Oct4 occupied sites after Oct4 knock-down.*

We agree that our study, and previous studies, lack the fine scale temporal resolution necessary to determine the kinetics with which altered chromatin accessibility, transcription factor binding and BRG1 chromatin occupancy occur following OCT4 removal. While we are immensely interested in these important questions, the current genetic ablation systems that are available are relatively slow (relying on natural turnover of pre-existing protein, ~24 hours for OCT4 and ~72 hours for BRG1) and asynchronous amongst the cell population. This presents a major technical barrier towards effectively assigning order of events in the type of kinetic experiments proposed by the reviewer. We envisage that addressing these important questions will require the development of rapid and synchronous protein ablations systems, likely degron based, and this remains an important challenge for future studies.

*5) The analyses of Oct4 or Brg1 regulated genes in Figure 4 do not give the impression that these factors regulate the same subset of genes in spite of the fact that very often they seem to bind to the same sites. How can this be explained?*

We thank the reviewer for raising this important point that we have now clarified with the support by several new analyses. In the absence of BRG1, our observations demonstrate that normal binding of pluripotency transcription factors OCT4, SOX2 and NANOG was disrupted at OCT4-bound distal regulatory elements, suggesting that this may affect expression of the pluripotency-associated transcriptional network. Somewhat surprisingly, previous work did not support a clear correlation between loss of BRG1 and the activity of OCT4 target genes (Ho et al., 2009, Ho et al., 2011, Zhang et al., 2014, Hainer et al., 2015), despite our observations that BRG1 plays an important role in transcription factor binding at many target sites associated with pluripotency-associated genes. However, we also observed that the effect on OCT4, SOX2 and NANOG binding following BRG1 removal varied in magnitude between individual sites with some sites actually displaying increases in transcription factor binding (Figure 5). We therefore reasoned that transcriptional effects following loss of BRG1 might not precisely correlate with the expression changes following OCT4 removal but may instead be related to alterations in OCT4, SOX2 and NANOG binding at individual sites that we observed in the BRG1-depleted cells. Based on this possibility we examined the relationship between gene expression changes following OCT4 and BRG1 ablation in more detail using unbiased clustering of OCT4-dependent genes based on their expression after removal of BRG1. We identified three separate types of responses; genes that showed reduced expression (28.1%), genes whose expression was unchanged (35.2%), and genes that had increased gene expression (36.7%) (Figure 5—figure supplement 3). Given that individual distal regulatory elements vary in the requirement for BRG1 in OCT4, SOX2 and NANOG binding, we examined whether expression of associated genes corresponded to the effects on transcription factor binding at associated distal regulatory elements. Importantly, this analysis revealed that the expression of genes associated with sites that rely on BRG1 for OCT4 binding tended to be reduced following deletion of BRG1 (Figure 5, Figure 5—figure supplement 4). In contrast, genes associated with unchanged or increased OCT4, SOX2 and NANOG binding tended to show increases in expression (Figure 5, Figure 5—figure supplement 4). Together these observations explain why previous studies have failed to identify a simple relationship between gene expression changes following OCT4 and BRG1 removal and demonstrates that reductions in expression of as subset of OCT4 target genes depends on loss of normal transcription factor binding following BRG1 removal. We now describe these points and additional analyses in the last paragraph of the subsection “Gene expression defects in the absence of BRG1 are linked to altered transcription factor binding”.

*Is it possible that Oct4 (or BRG1) are not regulating the analysed nearby genes?*

We used a simple approach based on coupling OCT4-bound distal regulatory elements with the closest TSS in order to examine changes in gene expression with altered chromatin accessibility and transcription factor binding. Therefore, this set of relationships will inevitably not represent the complete set of target genes affected by OCT4 bound distal regulatory elements and will also encompass false positives. However, in the absence of experimentally validated and high resolution genome-wide enhancer-promoter interaction maps we believe this is a reasonable approach that is likely to represent many functionally relevant relationships. Similar approaches have previously been used in genomic studies to broadly examine the function of distal regulatory elements in gene regulation (Kim et al., 2010, Degner et al., 2012).